# Liver microRNA transcriptome reveals miR-182 as link between type 2 diabetes and fatty liver disease in obesity

Christin Krause[1,2,3], Jan H Britsemmer[1,2,3], Miriam Bernecker[3,4], Anna Molenaar[3,4], Natalie Taege[1,2,3], Nuria Lopez-Alcantara[2,5], Cathleen Geißler[1,2], Meike Kaehler[6], Katharina Iben[1,2], Anna Judycka[1,2], Jonas Wagner[7], Stefan Wolter[7], Oliver Mann[7], Paul Pfluger[3,4,8], Ingolf Cascorbi[6], Hendrik Lehnert[2,3,9], Kerstin Stemmer[3,10], Sonja C Schriever[3,4]*, Henriette Kirchner[1,2,3]*

[1]Institute for Human Genetics, Division Epigenetics & Metabolism, University of Lübeck, Lübeck, Germany; [2]Center of Brain, Behaviour and Metabolism (CBBM), University of Lübeck, Lübeck, Germany; [3]German Center for Diabetes Research (DZD), Munich, Germany; [4]Research Unit NeuroBiology of Diabetes, Institute for Diabetes and Obesity, Helmholtz Centre, Munich, Germany; [5]Institute for Experimental Endocrinology, University of Lübeck, Lübeck, Germany; [6]Institute of Experimental and Clinical Pharmacology, University Hospital Schleswig-Holstein, Campus Kiel, Kiel, Germany; [7]Department of General, Visceral and Thoracic Surgery, University Medical Center Hamburg-Eppendorf, Hamburg, Germany; [8]Chair of Neurobiology of Diabetes, TUM School of Medicine, Technical University of Munich, Munich, Germany; [9]University Hospital of Coventry and Warwickshire, Coventry, United Kingdom; [10]Molecular Cell Biology, Institute of Theoretical Medicine, Faculty of Medicine, University of Augsburg, Augsburg, Germany

*For correspondence:
sonja.schriever@helmholtz-munich.de (SCS);
henriette.kirchner@uni-luebeck.de (HK)

Competing interest: The authors declare that no competing interests exist.

## Abstract

**Background:** The development of obesity-associated comorbidities such as type 2 diabetes (T2D) and hepatic steatosis has been linked to selected microRNAs in individual studies; however, an unbiased genome-wide approach to map T2D induced changes in the miRNAs landscape in human liver samples, and a subsequent robust identification and validation of target genes are still missing.

**Methods:** Liver biopsies from age- and gender-matched obese individuals with (n=20) or without (n=20) T2D were used for microRNA microarray analysis. The candidate microRNA and target genes were validated in 85 human liver samples, and subsequently mechanistically characterized in hepatic cells as well as by dietary interventions and hepatic overexpression in mice.

**Results:** Here, we present the human hepatic microRNA transcriptome of type 2 diabetes in liver biopsies and use a novel seed prediction tool to robustly identify microRNA target genes, which were then validated in a unique cohort of 85 human livers. Subsequent mouse studies identified a distinct signature of T2D-associated miRNAs, partly conserved in both species. Of those, human-murine miR-182–5 p was the most associated with whole-body glucose homeostasis and hepatic lipid metabolism. Its target gene *LRP6* was consistently lower expressed in livers of obese T2D humans and mice as well as under conditions of miR-182–5 p overexpression. Weight loss in obese mice decreased hepatic miR-182–5 p and restored *Lrp6* expression and other miR-182–5 p target genes. Hepatic overexpression of miR-182–5 p in mice rapidly decreased LRP6 protein levels and increased liver triglycerides and fasting insulin under obesogenic conditions after only seven days.

**Conclusions:** By mapping the hepatic miRNA-transcriptome of type 2 diabetic obese subjects, validating conserved miRNAs in diet-induced mice, and establishing a novel miRNA prediction tool, we

provide a robust and unique resource that will pave the way for future studies in the field. As proof of concept, we revealed that the repression of *LRP6* by miR-182–5 p, which promotes lipogenesis and impairs glucose homeostasis, provides a novel mechanistic link between T2D and non-alcoholic fatty liver disease, and demonstrate in vivo that miR-182–5 p can serve as a future drug target for the treatment of obesity-driven hepatic steatosis.

**Funding:** This work was supported by research funding from the Deutsche Forschungsgemeinschaft (KI 1887/2-1, KI 1887/2-2, KI 1887/3-1 and CRC-TR296), the European Research Council (ERC, CoG Yoyo LepReSens no. 101002247; PTP), the Helmholtz Association (Initiative and Networking Fund International Helmholtz Research School for Diabetes; MB) and the German Center for Diabetes Research (DZD Next Grant 82DZD09D1G).

## eLife assessment

Building on on the observation of an increase in miR-182-5p in diabetic patients, the authors investigated the role of miR-182-5p and its target gene LRP6 in dysregulated glucose tolerance and fatty acid metabolism in obese type 2 diabetics. The use of human livers complemented by supporting data in mice and cells are strengths, but the evidence presented remains **incomplete**. The findings provide **valuable** insights into the role of miRNAs in the regulation of liver metabolism and insulin sensitivity in individuals with diabetes and fatty liver disease.

## Introduction

The global prevalence of obesity has drastically increased over the past years and is associated with numerous metabolic and psychological diseases (*Blüher, 2019*). Accordingly, obesity is the leading risk factor for T2D and non-alcoholic fatty liver diseases (NAFLD). Yet, while almost 90% of individuals with T2D are overweight or obese, not all obese individuals develop T2D (*Narayan et al., 2007*). Multiple factors play a role in the development of T2D, including genetics, lifestyle, and the gut microbiome (*Chobot et al., 2018*). Non-genetic risk factors such as epigenetic changes induced by aging, weight gain, and intrauterine programming were shown to contribute to the increasing prevalence of T2D (*Ling and Groop, 2009*). microRNAs (miRNAs), as one epigenetic mechanism, have the potential to simultaneously regulate the expression of multiple genes and entire pathways (*Friedman et al., 2009*). Additionally, miRNAs can mediate organ cross-talk (*Gurwitz, 2015*), and the dysregulation of a few miRNAs can have deleterious effects on a systemic level (reviewed in *Ambros, 2001*). Aberrant expression of miRNAs in T2-diabetic subjects was already shown in human adipose tissue, pancreatic beta cells, skeletal muscle, and blood (reviewed in *Agbu and Carthew, 2021*). However, although the liver is the central organ to maintain whole-body glucose homeostasis (*Rines et al., 2016*) little is known about alterations of hepatic miRNA expression in T2-diabetic subjects.

  miR-103 and miR-107 directly regulate insulin sensitivity in liver and adipose tissue of obese mice by repressing caveolin-1 and insulin receptor signal transduction (*Trajkovski et al., 2011*). Likewise, miR-802 impairs hepatic insulin sensitivity (*Kornfeld et al., 2013*) and the highly expressed miRNAs miR-34a (*Ding et al., 2015*) and miR-122 (*Yang et al., 2012*) perturb hepatic glucose and lipid metabolism in mice. Nevertheless, most studies on hepatic miRNAs in T2D pathogenesis were performed in laboratory rodents (*Trajkovski et al., 2011*; *Li et al., 2009*; *Yang et al., 2015*; *Yang et al., 2016*) or liver cell lines (*Gerin et al., 2010*), and human data is largely elusive. We have shown that miR-let-7e is increased in livers of T2-diabetic subjects and associated with down-regulation of the insulin receptor substrate 2 gene (*IRS2*) (*Krause et al., 2020*). A comprehensive evaluation of all hepatic miRNAs transcribed and dysregulated in T2-diabetic subjects is nonetheless missing. Importantly, miRNAs are already used as drug-targets for cancer treatments and provide an excellent basis to develop novel therapies for metabolic diseases (*Krützfeldt et al., 2005*). Accordingly, here we aimed to conduct a thorough profiling of hepatic miRNA signatures that are associated with T2D in obese human subjects with or without type 2 diabetes, to identify miRNAs that control key genes involved in metabolic pathways as well as NAFLD and hepatic insulin resistance etiology.

## Methods

### Study design human cohort

All participants of the human cohort signed informed consent. The study was approved by the local ethics committee (Ärztekammer Hamburg, approval ID PV4889) and conformed to the ethical guidelines of the 1975 Declaration of Helsinki.

Liver biopsies were obtained during bariatric surgery at the University Hospital Eppendorf (UKE, Hamburg) as previously described (*Bartel, 2009*). Blood samples were drawn on the day of surgery after 6 hr fasting. All clinical parameters were measured at the Institut für Klinische Chemie und Laboratoriumsmedizin, Zentrum für Diagnostik, Universitätsklinikum Eppendorf, Hamburg, Germany, according to the DIN EN ISO 15189:2014 certification. Serum glucose, cholesterol, HDL, and triacyl-glycerols were determined using photometric assays (kinetic bichromatic analysis; in-house assay). HbA1c was quantified by capillary electrophoresis or by turbidimetric inhibition assays (in-house assay). Insulin levels were quantified from frozen serum samples using the Human/Canine/Porcine Insulin Quantikine ELISA Kit (R&D Systems, Inc, Minneapolis, US). Subjects were categorized by (1) the use of anti-diabetic medication and (2) by definition of the American Diabetes Association (ADA) into subjects with T2D (receiving anti-diabetes medication or HbA1c$\geq$6.5%, n=41) and non-diabetic controls (not receiving anti-diabetes medication and HbA1c<5.7%; n=44, *Supplementary file 1a*). Anti-diabetic treatment was not uniform in the T2D group and included oral glucose-lowering medications or insulin. The mean BMI was 52.37 kg/m$^2$ ranging from 32.19 to 84.87 kg/m$^2$. The complete cohort was imbalanced in age, which was treated as a potential confounding factor and adjusted for if applicable.

The microarray sub-cohort n=20 T2-diabetic and n=20 non-diabetic control subjects, *Supplementary file 1a* was matched for age, sex and BMI, and was part of the complete study cohort. Both disease groups contained individuals with different stages of NAFLD determined as NAFLD activity score (NAS) ranging from 0 (no NAFL) to 7 (high degree of fibrosis and steatosis), which was considered during data analysis. The NAFLD activity score and degree of hepatic steatosis were determined during surgery according to the current recommendations by two expert pathologists. The sectional reporting guidelines were used according to STROBE.

### Study design mouse cohorts

The murine studies were approved by the State of Bavaria, Germany (approval ID ROB-55.2–2532.Vet_02-19-167). All experiments were performed in adult male C57BL/6 J mice purchased from Janvier Labs (Saint-Berthevin, Cedex, France). Mice were maintained on a 12 hr light-dark cycle with free access to water and a standard chow diet (Altromin, #1314) or 58% high-fat diet (HFD) (Research Diets, D12331). For the weight cycling study, mice were subjected to HFD for 24 weeks before intervention: The group termed HFD remained on HFD during the study. The remaining two groups of DIO mice were divided as follows: Diet switch (HC) animals continued to receive ad libitum access to HFD for 12 weeks before they were switched to a chow diet for the remaining 12 weeks of the study. In contrast, weight cycling (YoYo) mice were switched to a chow diet on day 0 of the study for 12 weeks and then fed again ad libitum with HFD for another 12 weeks. Age-matched mice fed chow were used as a control group. At the end of the diet-intervention study, all mice were fasted for 4 hr at the start of the light phase before being sacrificed by cervical dislocation for organ withdrawal. To obtain plasma blood was collected in EDTA-containing tubes and centrifuged at 4°C and 2000 × g for 10 min. Mice used for the DIO miRNA microarray analysis were fed HFD for 28 weeks. Hepatic triglyceride content was determined using the triglyceride assay kit (Wako Chemicals, Neuss, Germany). Plasma leptin and insulin levels were measured by murine ELISA kits (Merck Millipore, Darmstadt, Germany, Leptin: EZML-82K, Insulin: EZRMI-13K). Group sizes were based on power analysis.

### Gene expression in the human liver

Whole cell RNA was extracted from 25 mg of snap-frozen liver using the miRNeasy mini kit (QIAGEN, Hilden, Germany) and quantified spectrometrically. 2 µg of RNA was reverse transcribed into cDNA using the SuperScript VILO cDNA synthesis kit (Invitrogen, Carlsbad, US). Gene expression was measured by qPCR in duplicates and calculated with the ΔΔCt method. Data were normalized to CASC3 expression as reference gene (*Krause et al., 2020*). Primer sequences and assay-IDs are given in *Supplementary file 1b and c*.

## Array-based miRNA transcriptome analysis

500 ng RNA from 40 subjects (n=20 type-2-diabetic, n=20 non-diabetic) and 1000 ng RNA from n=6 mice kept on HFD for 28 weeks were used for probe preparation by the FlashTagTM Biotin HSR RNA Labelling Kit (Applied Biosystems, Foster City, US). Prepared samples and control oligo B2 from the GeneChipTM Hybridization Control Kit (Applied Biosystems, Foster City, US) were hybridized to the GeneChipTM miRNA 4.0 Array (Applied Biosystems, Foster City, US). After incubation for 18 hr and washing, each array was scanned by a GeneChipTM Scanner 3000 (Applied Biosystems, Foster City, US).

## Gene expression analysis of murine liver samples

Whole cell RNA was extracted from 20 mg of snap-frozen liver using the miRNeasy mini kit (QIAGEN, Hilden, Germany) and quantified spectrometrically. 1 µg of RNA was reverse transcribed using the QuantiTect Reverse Transcription kit (QIAGEN, Hilden, Germany). Expression of Acaca, Dhcr7, Elovl6, Fasn, Foxo1, G6pc, Hmgcs1, Insig1, Insr, Irs1, Scd1, Lrp6, Pck1, Pdk1, Pnpla2, and Ppara was measured in duplicates by SYBR green qPCR (SYBR Green Master Mix, Thermo Fisher Scientific, Inc, Rockford, US) and calculated with the ΔΔCt method using Hprt1 as the housekeeping gene. Primer sequences are listed in *Supplementary file 1b*.

## microRNA expression analysis in human and murine liver

1 µg total human or mouse liver RNA was reversely transcribed with the qScript microRNA cDNA synthesis kit (Quanta Bioscience, Beverly, US). Self-designed forward and a pre-designed universal reverse primers were used for the expression measurement of miR-182–5 p by SYBR green qPCR (PerfeCTa SYBR Green SuperMix, Quanta Bioscience, Beverly, US) in duplicates. Data were analyzed with the ΔΔCt method using hsa-miR-24–3 p as a housekeeping gene (*Supplementary file 1b*).

## miRNA isolation and miRNA expression analysis in human serum

Serum miRNA was extracted from 200 µl frozen serum by the miRNeasy Serum/Plasma Advanced kit (QIAGEN, Hilden, Germany) using the spike-in control cel-miR-39–3 p (QIAGEN, Hilden, Germany). MiRNAs from 5 µl eluate (corresponding to 40 µl serum input) were reverse transcribed by qScript microRNA cDNA synthesis kit (Quanta Biosciences, Beverly, US). Expression was quantified as mentioned before. Each qPCR reaction contained 266 nl input serum. Target expression was normalized to the spike-in control (cel-miR-39–3 p 5'-CACCGGGTGTAAATCAGCTTG) expression using the ΔΔCt method.

## Plasmid cloning, site-directed mutagenesis, and luciferase reporter gene assay

The 3'-UTR of *LRP6* containing a potential 7mer-m8 recognition site was generated by PCR amplification from human hepatic cDNA with primer containing the recognition motifs for NotI and XhoI restriction. (*Supplementary file 1b*). After restriction digestion with *NotI* and *XhoI* (NEB, Ipswich, US) the PCR fragment was directionally cloned downstream of the Renilla luciferase ORF in the psiCHECK-2 vector (Promega, Madison, US) using the Quick Ligase kit (M2200, NEB, Ipswich, US). A constitutively expressed Firefly luciferase gene on the same vector served for normalization of the Renilla luciferase signal. 10 ng of this plasmid served as PCR template for a site-directed mutagenesis whereby the central nucleotide of the seed sequence (5'-TTGGCAA) was exchanged by an adenosine to (5'-TTGACAA). The site-directed mutagenesis reaction was performed by using the Phusion Hot Start II High-Fidelity PCR Master Mix (Thermo Scientific, Waltham, US) and a PCR protocol for 12 cycles. The parental strand was digested by *DpnI* for 15 min (NEB, Ipswich, US). Plasmids were transformed in DH5α *E. coli* bacteria (NEB, Ipswich, US) and extracted with the QIAprep Spin Midiprep kit (QIAGEN, Hilden, Germany). HEK-293 cells were co-transfected in triplicates with miRNA precursor mimics (pri-miRNA-182–5 p PM12369 or negative control #1 AM17110, Ambion, Applied Biosystems, Foster City, US) and the psiCHECK-2 vector containing the 3'UTR with either the consensus seed or the mutated seed sequence. Cells were harvested after 48 hr. Renilla and Firefly luciferase signals were measured using the Dual-Glo luciferase reporter gene assay (Promega, Madison, US). Each experiment was performed three times. HEK-293 cells (obtained from ATCC, Manassas, US,

Mycoplasma-free) were cultivated at 37 °C and 5% CO2 in high glucose (4.5 g/l) DMEM medium supplemented with 10% (vol./vol.) FBS and 1% penicillin/streptomycin.

HEK-293 cells were co-transfected in triplicates with miRNA precursor mimics (pri-miRNA-182–5 p PM12369 and negative control #1 AM17110, Ambion, Applied Biosystems, Foster City, US) and the luciferase plasmid containing the 3'UTR with either the consensus seed or the mutated seed sequence generated by site-directed mutagenesis using a final concentration of 10 nmol/l for mimic or negative control and 50 ng plasmid per well of a 96-well plate. Transfection was performed by reverse transfection using LipofectamineTM 3000 (Invitrogen, Carlsbad, US) and a cell concentration of 100.000 cells per 1 ml high glucose (4.5 g/l) DMEM medium supplemented with 10% (vol./vol.) FBS. Cells were harvested after 48 hr of incubation and lysed in 80 µl 1 x passive lysis buffer (Dual-Glo luciferase reporter gene assay, Promega, Madison, US). Renilla and Firefly luciferase signals were measured in 20 µl cell lysates by using the Dual-Glo luciferase reporter gene assay (Promega, Madison, US). The Renilla signal was normalized to the Firefly signal which are both encoded on the same plasmid. The miR182-5p mimic signal was compared to the respective negative control in each experiment. Each experiment was performed three times.

## miRNA overexpression in HepG2 cells

HepG2 (obtained from ATCC, Manassas, US, Mycoplasma-free) were cultivated at 37 °C and 5% CO2 in low glucose (1.5 g/l) DMEM medium supplemented with 10% (vol./vol.) FBS, 1 mM sodium pyruvate, and 1% penicillin/streptomycin. miRNA precursor mimics as mentioned before were reverse transfected into HepG2 cells in triplicates with a final concentration of 10 nmol/l per well of a 6-well plate using Lipofectamine RNAiMAX Transfection Reagent. Transfected cells were washed with PBS and lysed in 700 µl TRIzol after 48 hr incubation. Whole cell RNA was isolated by miRNeasy mini kit (QIAGEN, Hilden, Germany). The quantification of HepG2 miR-182–5 p expression was performed by reverse transcription of 10 ng of HepG2 whole cell RNA with the TaqManTM Advanced cDNA synthesis kit (Applied Biosystems, Foster City, US) using miR-24–3 p as housekeeper assay. For the quantification of potential target genes, 500 ng RNA was reversely transcribed into cDNA using the High-Capacity cDNA Reverse Transcription Kit (Applied Biosystems, Foster City, US). Gene expression measurement was performed as previously stated for hepatic genes. Negative control #1 transfected cells served as a reference for comparison. For western blot analysis of the LRP6 protein, cells were harvested in ice-cold PBS after 72 h incubation. Total protein was isolated by incubation of the cell pellet for 30 min on ice in modified RIPA buffer (50 mmol/l Tris-HCl, 1% (vol./vol.) NP-40, 0.25% (wt/vol.) Na-deoxycholat, 150 mmol/l NaCl, 1 mmol/l EDTA, 1 mmol/l PMSF, Na3VO4, 1 mmol/l NaF, supplemented with protease inhibitor cocktail cOmpleteTM (Roche, Basel, CH)), subsequent centrifugation for 10 min at 14.000 g and 4 °C and the supernatant was collected. For western blot analysis for the measurement of phospho-Akt, transfected cells were treated with 20 nmol/l insulin (Lantus 100 E/ml, Sanofi, Paris, FR) for 10 min prior to harvest. Each experiment was performed three times.

## Western blot analysis

25 µg (Akt/pAkt) to 35 µg (LRP6) of HepG2 protein lysate was separated for 1.5 hr at 100 V by SDS-Page (7.5% (wt/vol.) acrylamide, TGX Stain-FreeTM FastCastTM Acrylamide Kit, Bio-Rad Laboratories, Inc, Hercules, US) and blotted on nitrocellulose membranes by fast blotting (mixed MW, Trans-Blot TurboTM, Bio-Rad Laboratories, Inc, Hercules, US). For the analysis of LRP6 in murine liver, protein was extracted by homogenization in modified RIPA buffer and subsequent centrifugation for 20 min at 18,000 g and 4 °C. 20 µg protein lysate was separated by SDS-Page (7.5% (wt/vol.) acrylamide Criterion TGX Precast Midi Protein gel, Bio-Rad Laboratories Inc, US) and blotted on 0.2 µm PVDF membranes (Bio-Rad Laboratories Inc, US). Total protein was imaged from the membrane as a control after UV-activation of the gel for 60 s. Unspecific binding sites were blocked by incubation for 1 hr at room temperature in 5% (wt/vol.) milk in TBS + 0.1% (vol./vol.) Tween (TBST). The blots were incubated with primary antibodies against LRP6 (1:1000, EPR2423(2) ab134146 rabbit mAb, lot GR3256666-1, Abcam, Cambridge, UK) or HSP90 (1:1000, C45G5 rabbit mAb, lot 5, Cell Signalling Technology, Danvers, US) over night at 4 °C. After washing in TBST, the membrane was incubated for 1 hr at room temperature with a secondary antibody (1:5000) conjugated with HRP (polyclonal goat anti-rabbit HRP, lot 20061231, Dako, Agilent, Santa Clara, US). For the analysis of phospho-Akt/Akt protein, the protein blots were first incubated with a primary antibody against phospho-Akt Ser473

(1:1000, D9E XP rabbit mAb #4060, lot 23, Cell Signaling Technology, Danvers, US) and after stripping of the membrane with a primary antibody against total Akt (1:1000, rabbit pAb #9272, lot 28, Cell Signaling Technology, Danvers, US). Each experiment was performed three times.

## Glucose uptake of HepG2 cells

miR-182–5 p precursor mimics, anti-miR miRNA inhibitors (AM10801, Ambion, Applied Biosystems, Foster City, US) and corresponding controls were reverse transfected in HepG2 cells in triplicates as described above. A glucose uptake assay (Glucose Uptake-GloTM Assay, Promega, Madison, US) was performed in 6-well plates using a final concentration of 1 mM 2-deoxyglucose (2DG) in PBS for the uptake reaction. Prior to incubation with 2DG, the cells were treated for 10 min with 20 nM insulin (Lantus 100 E/ml, Sanofi, Paris, FR). The luciferase signal was measured in duplicates for each sample in a 96-well format. The mimic signal was normalized to the respective cells treated with negative control. Each experiment was performed three times.

## In vivo overexpression of miR-182-5p in mouse liver

Male C57BL/6 J mice were purchased at 5–6 weeks of age from Janvier Labs (Saint-Berthevin, Cedex, France), maintained as described above, and fed with HFD for four weeks prior and throughout the study. Based on their body weight, the mice were assigned at day –1 to either the control or the miR-182–5 p mimic group so that each group contained eight weight-matched mice. On day 0, the first injection of 1 mg miR-182–5 p-mimic or negative control per kg body weight was performed via tail vein using Invivofectamine 3.0 (Invitrogen, Carlsbad, US) as described by the manufacturer. A second injection was performed at day 3.5 to maintain the microRNA level. Mice were phenotypically characterized by NMR on days –1 and 7 and a glucose tolerance test after 6 h fasting on day 5. Plasma was collected from blood samples during the GTT to determine insulin levels by ELISA (EZRMI-13K, EMD Millipore Corporation, Burlington, US). All mice were sacrificed at day 7 after a 3 hr fast. Liver, spleen, and kidney were collected and snap frozen for the extraction of RNA and protein as described before. Hepatic triglycerides were measured using the Triglyceride Quantification colorimetric-/fluorometric Kit (MAK266, Sigma-Aldrich, St. Louis, US) from 50 mg of liquid-nitrogen pulverized tissue homogenized in 5% (wt/vol.) IGEPAL CA-630 (Sigma-Aldrich, St. Louis, US). A hematoxylin and eosin (HE) staining was performed from sections of the fresh frozen liver for visualization of lipid accumulation.

## Microarray statistics

Regression analysis, statistical tests, and visualization was performed by MATLAB R2020a (The MathWorks, Natick, US), R 3.5.1 (The R Foundation for Statistical Computing, Vienna, Austria), and GraphPad Prism 7.05 (GraphPad Software, Inc, San Diego, US). Generated CEL data was imported into Transcriptome Analysis Console (TAC) 4.0 and pre-processed for further analysis in MATLAB R2020a and R 3.5.1. Array data was normalized by Robust Multi-chip Analysis (RMA) algorithm as indicated by the manufacturer, which contains background adjustment, log2 transformation, and quantile normalization to increase the fold change ratios. Probesets were considered expressed if more than 50% of probes have a significantly ($p < 0.05$) higher detection signal than the background (DABG, detection above background).

Linear regression models for continuous responses (metabolic traits) or logistic regression models for the incidence of T2D were generated in MATLAB. Consequently, age, sex, BMI, and the NAFLD activity score (NAS) as marker for hepatic steatosis and fibrosis were used as additional cofactors if not used as response variable:

> Trait or T2D = $\beta_0 + \beta_1 \ast NAS + \beta_2 \ast age + \beta_3 \ast sex + \beta_4 \ast BMI + \beta_5 \ast log2miRNA$
> By using NAS as a cofactor we ensured that NAFLD is not driving the altered miRNA expression in T2D.
> To control for associations with confounding factors (age, BMI) also the HbA1c level was considered as a cofactor:
> Age or BMI = $\beta_0 + \beta_1 \ast HbA1c + \beta_3 \ast sex + \beta_4 \ast [BMI \ or \ age] + \beta_5 \ast log2miRNA$
> Resulting effect sizes β describe the change of the trait if the miRNA expression changes by 1 log2 value.

A p-value <0.05 was considered associated. Since FDR adjustment of thereby generated q-values resulted in no significantly associated miRNA (q<0.05), another approach for candidate identification was used. First, we filtered for confidently expressed microRNAs by applying a log2 threshold of 2.3. This is reasoned by the minimal group mean log2 value which was declared as 'is expressed' by the TAC software and which could be quantified by qPCR. Moreover, multiple associations to the incidence of type 2 diabetes or other metabolic traits and missing associations to BMI and age were mandatory to be declared as candidate genes for qPCR validation. Second, we evaluated which microRNAs are conserved between human and mouse by 7mer seed match analysis. By use of these exclusion criteria, we were able to identify candidate miRNAs which are differentially expressed in T2D independently of obesity and not driven by NAS.

## General statistics

All data were adjusted for multiple testing (FDR) according to Benjamini-Hochberg when more than one test was performed. Changes in gene expression, luciferase signal, glucose uptake, protein abundance, and metabolic parameters between two groups were tested by student's t-test and between multiple groups by One-way ANOVA. All ΔCt-values which were not within a three standard deviations interval of all samples for the respective gene were defined as outliers and excluded for further analysis. Normal distribution was tested using the Lilliefors test implemented in MATLAB with a significance level of p<0.05. ΔCt values were correlated with metabolic parameters and other genes by Pearson correlation and corrected for age and gender by linear regression if applicable. Results prior adjustment are indicated with a p-value. A q-value <0.05 was considered as significant. Correlation matrices were plotted using the ggcorr-function of ggplot2-extension GGally (https://CRAN.R-project.org/package=GGally) and R 3.5.1 (The R Foundation for Statistical Computing, Vienna, Austria). Only applicable associations analyzed by Pearson's correlation are plotted. Non-tested associations are indicated by gray squares. Fold-change heatmaps and other graphs were generated by using GraphPad Prism 7.05 (GraphPad Software, Inc, San Diego, US).

## Target gene and pathway analysis

Target genes of miRNAs were taken from miRTarBase, and TarBase (*Chou et al., 2018*; *Karagkouni et al., 2018*) and integrated into a SQLite database and queried by a function we termed 'miRNA-Nvis-tool' implemented in MATLAB R2020a (The MathWorks, Natick, US). We implemented a target gene prediction tool as part of the microarray analysis framework to identify and list potential target genes of miRNAs. Thereby miRNA Nvis uses the mature miRNA and the respective 3'-UTR sequence of the target genes as a string and performs a seed match (2–7 nt of the 5' miRNA sequence string) for different pairing modes (8mer, 7mer-m8, 7mer-A1, 6mer and offset 6mer) (*Bartel, 2009*) and computes favorable conditions for seed binding, such as the AU content within 60 nt surrounding the seed, additional binding 3' to the seed sequence of the miRNA, the relative position of the seed within the 3'-UTR of the potential target, proximity to the stop codon and additional or cooperative binding of the same or other miRNA species (*Bartel, 2009*; *Peterson et al., 2014*). The output can be exported as a table with all information on binding conditions and position for all pairing options within the sequence input string. Target genes were then manually selected according to literature research and after stratification for T2D-related gene ontology terms. For pathway enrichment analysis, we used only experimentally validated target genes from miRTarBase (*Chou et al., 2018*) as input for enrichKEGG from clusterProfiler (version 3.0.2 and R version 3.5) with the options organism = 'hsa,' keyType = 'kegg,' minGSize = 1 and otherwise default parameters.

## Results

### Type 2 diabetes in obese individuals is associated with 28 miRNAs

To identify hepatic miRNAs that are associated with T2D independently of obesity we performed an array-based miRNA transcriptome analysis in liver biopsies of obese subjects with (n=20) and without T2D (n=20; *Supplementary file 1a*). The array contained a total of 4603 human pre-miRNAs and mature miRNAs of which only 694 mature miRNAs (27% of the total array-based miRNAs) were expressed in obese livers, regardless of the presence of T2D (*Figure 1A*). To increase the confidence of detection, the minimum signal threshold was set to a log2 value >2.3 resulting in the detection

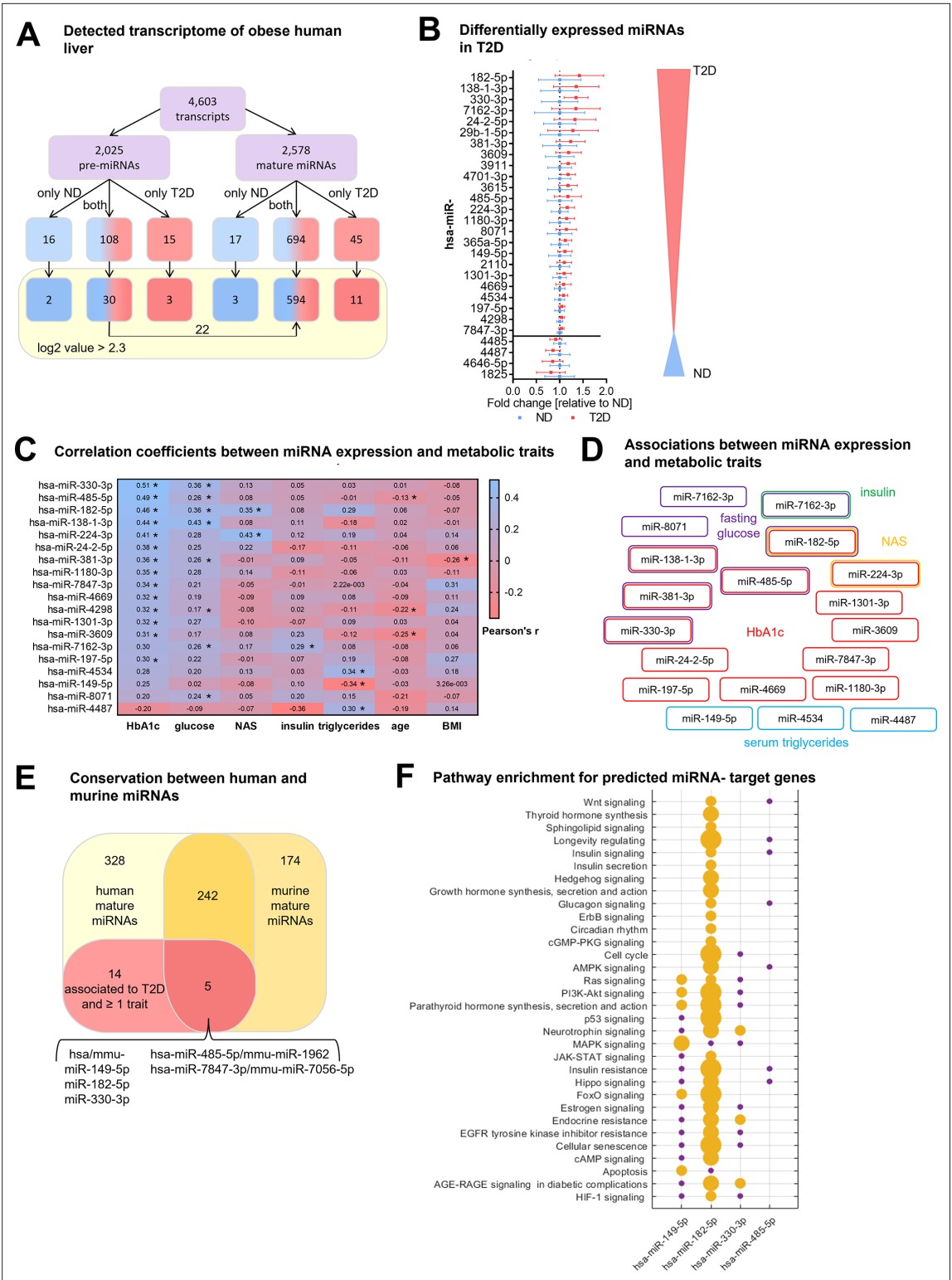

**Figure 1.** Type 2 diabetes in obese individuals is associated with a distinct hepatic miRNA signature. (**A**) Overview of all analyzed human transcripts within the GeneChip miR4.0 assay. (**B**) Expression of the 28 regulated miRNAs in type 2 diabetes (T2D, n=20) compared to non-diabetic (ND, n=20) after adjustment for age, sex, BMI, and NAFLD activity score (NAS) in logistic regression models ranked by fold-change. Data is represented as relative mean to ND ± SD. (**C**) Pearson's correlation coefficients (**r**) for the correlation between microarray miRNA log2 values and metabolic parameters (n=40). miRNAs are ranked by their association strength to first HbA1c, glucose, NAS, insulin, triglycerides, age, and lastly BMI. (**D**) Mapping of significant correlations from (**C**) to metabolic parameters. (**E**) Venn diagram of all mature miRNA detected by microarray measurement in human and diet-induced

*Figure 1 continued on next page*

*Figure 1 continued*

obese (DIO) murine liver samples. (**F**) Pathway enrichment analysis of validated target genes of conserved miRNAs from (**E**) available in miRTarBase (19). Yellow dots indicate a significant enrichment of target genes in the respective pathway which correlates with the dot size and violet dots a potential implication meaning a non-significant enrichment of target genes in this pathway. Corrected for multiple testing: *q<0.05.

of 594 mature miRNAs (23%) with high confidence in the obese human liver samples of both groups (*Figure 1A*). Of the 30 pre-miRNAs that passed the minimal expression threshold 22 pre-miRNAs (73%) were also detected as mature miRNA transcripts indicating that they undergo a full maturation process in obese human livers (*Figure 1*).

To identify hepatic miRNAs that potentially drive T2D pathogenesis in obesity we applied a logistic regression analysis for the 594 mature miRNAs using the NAFLD activity score (NAS) as cofactor to exclude any bias by hepatic fat content, lobular inflammation, and fibrosis. 28 mature miRNAs were differentially expressed in T2D subjects compared to obese non-diabetic (ND) controls of which 24 miRNAs were up and four miRNAs were down regulated (*Figure 1B* and *Supplementary file 1d*). The expression levels of 19 of 28 differentially expressed miRNAs correlated significantly with at least one metabolic trait such as HbA1c, fasting blood glucose, and NAS (*Figure 1C*, *Supplementary file 1d*). miRNAs that were significantly associated with age or BMI were excluded (*Supplementary file 1e*) since we aimed to uncover miRNAs that contribute to T2D in obese subjects irrespective of age and BMI. Six miRNAs were associated with two metabolic traits and miR-182–5 p was the only miRNA associated with three metabolic traits: HbA1c, fasting glucose, and NAS (*Figure 1D*).

Next, we analyzed livers of DIO, insulin-resistant male C57BL/6 J mice and compared the murine array-based hepatic miRNA expression with the human obese hepatic miRNA transcriptome to identify miRNAs that are conserved between species. Of the 594 mature human miRNAs detected with high confidence, 247 miRNAs (42%) were also expressed in livers of DIO mice based on a seed sequence match between human and murine mature miRNAs (*Figure 1E*). The remaining 342 miRNAs were exclusively expressed in humans. Similarly, about half of the murine hepatic miRNAs (n=174) were either not conserved or not expressed in human liver. Only five out of 19 miRNAs that are dysregulated in liver of obese T2D subjects and are associated with at least one metabolic trait (HbA1c, fasting glucose, insulin, NAS, serum triglycerides) after adjustment for confounding factors (age, sex, BMI, and NAS) are conserved in mice (*Figure 1E*, *Supplementary file 1f*). Among these conserved miRNAs is miR-182–5 p.

Many of the T2D-associated miRNAs identified in our human liver (*Figure 1C*) were annotated only recently, and their target genes remain to be validated and integrated into public databases. However, for 4 out of the 5 conserved T2D-associated miRNAs information about potential target genes is available in miRTarBase (*Peterson et al., 2014*). These miRNAs and their target genes are significantly involved in metabolic pathways including insulin signaling, insulin resistance,f and PI3K-Akt signaling ($p_{adj}$ <0.05, *Figure 1F* and *Supplementary file 1g*). Taken together, our data indicate that the identified hepatic miRNAs associate with T2D pathogenesis in human and murine obesity.

## hsa-miRNA-182-5p is a gate keeper for *LRP6*-dependent regulation of hepatic glucose and lipid metabolism

Prompted by the strong association of hepatic miR-182–5 p to metabolic traits and its likely regulatory impact on genes involved in glucose homeostasis and lipid metabolism, we next extended our, now qPCR-based, expression analysis of miR-182–5 p to liver biopsies of our full study cohort of n=85 obese subjects of which n=44 were ND and n=41 were T2D according to ADA stratification and use of anti-diabetic medication. The full cohort included the original 40 discovery individuals (*Supplementary file 1a*). Confirming our microarray results, the expression of miR-182–5 p was 2.3-fold upregulated in the diabetic livers (*Figure 2A*). In this full cohort hepatic miR-182–5 p expression was significantly correlated with age after adjustment for multiple testing (*Figure 2B*, blue box), therefore, we adjusted all consecutive regression analysis for age. The correlations of hepatic miR-182–5 p expression with HbA1c and NAS were confirmed in the full cohort as well as a significant association with fasting glucose levels (*Figure 2B* red box). Furthermore, miR-182–5 p correlated significantly with serum triglyceride levels and hepatic lipid content in the complete cohort, both important indicators of lipid homeostasis (*Figure 2B*, red box). These data confirm that, miR-182–5 p is potentially involved in the regulation of glucose tolerance and lipid metabolism in obese subjects. To test whether cell-free

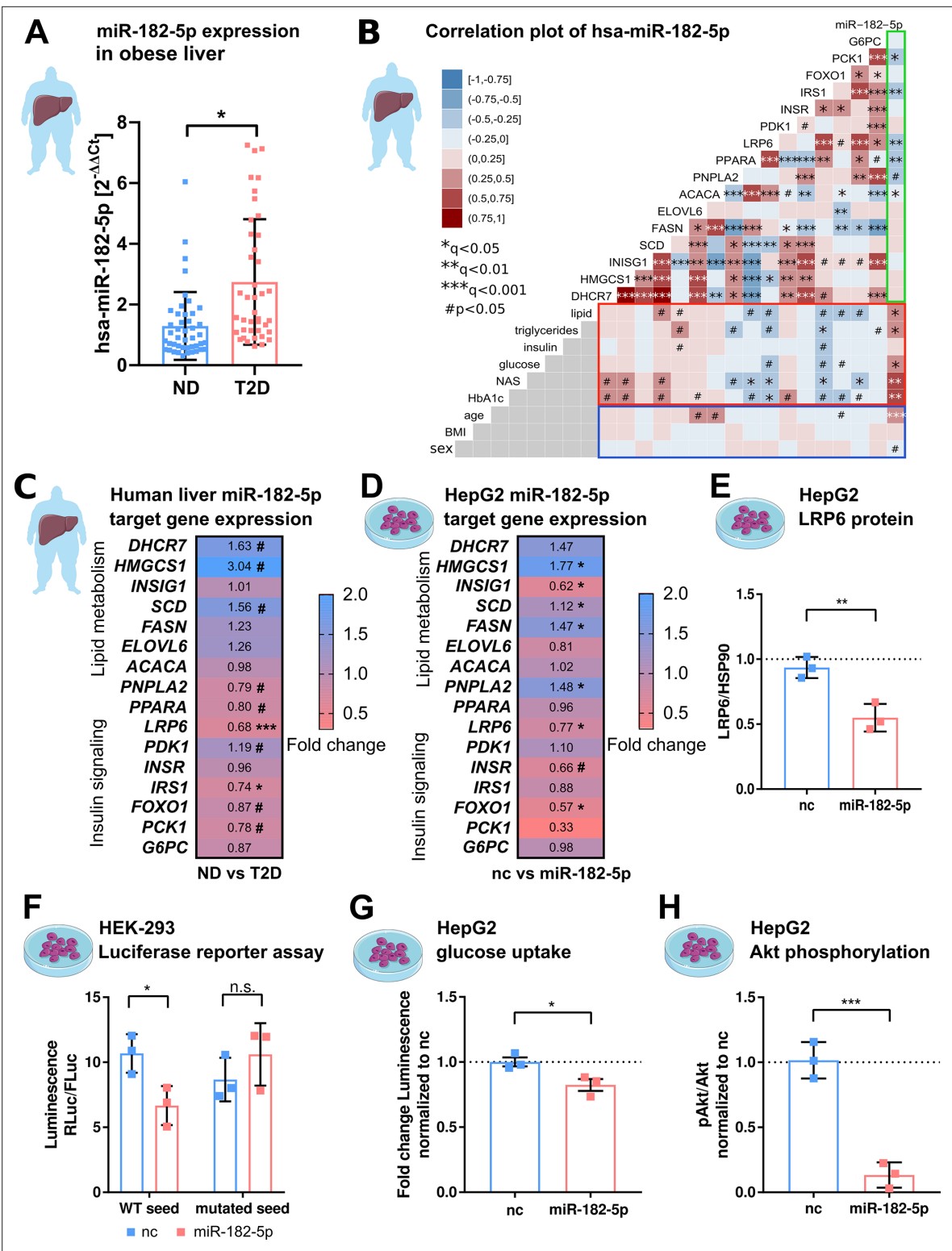

**Figure 2.** hsa-miRNA-182–5 p is a gate keeper for LRP6-dependent regulation of glucose homeostasis and hepatic lipid metabolism. (**A**) Expression of miR-182–5 p is 2.3-fold upregulated in liver tissue of obese subjects with type 2 diabetes (T2D, HbA1c≥6.5% or anti-diabetic medication) compared to non-diabetic (ND, HbA1c<5.7 %) obese subjects in the extended human liver cohort (n=85). (**B**) Correlation plot of hepatic miR-182–5 p expression and its target genes (*ACACA*: n=85, *DHCR7*: n=82, *ELOVL6*: n=83, *FASN*: n=84, *FOXO1*: n=80, *G6PC*: n=84, *HMGCS1*: n=77, *INSIG1*: n=84, *INSR*: n=80, *IRS1*: n=78, *LRP6*: n=80, *PCK1*: n=84, *PDK1*: n=84, *PNPLA2*: n=77, *SCD*: n=80) in human liver (green box) and of gene expression with metabolic parameters from blood (red box) as well as with confounders (blue box, all n=85). Non-tested correlations are indicated by gray squares. (**C**) Expression

*Figure 2 continued on next page*

*Figure 2 continued*

of miR-182–5 p target genes in human diabetic liver (see also *Figure 2—figure supplement 1C*; same n as indicated in B). (**D**) Expression of target genes after overexpressing miR-182–5 p for 48 h in HepG2 cells in comparison to a negative control (nc, n=3). (**E**) Protein abundance of the novel target gene *LRP6* is reduced after overexpression of miR-182–5 p for 72 h in HepG2 cells (n=3). (**F**) Overexpression of miR-182–5 p in HEK-293 cells decreases luciferase activity after 48 h in a luciferase reporter assay for the LRP6 wild-type (WT) sequence but not in the mutated seed (n=3). (**G**) Glucose uptake is significantly reduced (0.82-fold) in HepG2 cells after 48 h of miR-182–5 p overexpression and acute insulin stimulation for 20 min (n=3). (**H**) Measurement of pAkt/Akt via Western blot indicates insulin resistance in HepG2 cells 48 hr after overexpression of miR-182–5 p (0.13-fold, n=3). Data are shown as scatter dot plots with mean ± SD (**A, E–H**) or correlation matrices (**B–D**). Without multiple testing correction: ***p<0.001, **p<0.01, *p<0.05 (**A, E–H**); corrected for multiple testing: ***q<0.001, *q<0.05, significant prior to adjustment: #p<0.05 (**B,C,D**); Students t-test (**A,C,D, E–H**) or Pearson's correlation (**B**).

The online version of this article includes the following source data and figure supplement(s) for figure 2:

**Source data 1.** Original file for the Western blot analysis of HepG2 LRP6 and HSP90 after mimic overexpression.

**Source data 2.** PDF containing *Figure 2E* highlighted bands and sample labels for protein quantification.

**Source data 3.** Original file for the Western blot analysis of HepG2 Akt and phosphor-Akt after mimic overexpression.

**Source data 4.** PDF containing *Figure 2H* highlighted bands and sample labels for protein quantification.

**Source data 5.** Phenotypic and qPCR expression (dCT values) data of the human cohort.

**Figure supplement 1.** Hepatic gene expression of miR-182–5 p target genes and correlation to miRNA expression as described by correlation plot (main *Figure 2B*) and heat map (main *Figure 2C*).

**Figure supplement 2.** Mimic transfection in HepG2 cells led to a significant 400-fold upregulation of miR-182–5 p (Students t-test, n=3).

**Figure supplement 3.** Western blot lanes used for quantification and which are corresponding to *Figure 2E* (**A**) or *Figure 2H* (**B**).

**Figure supplement 4.** Effects of suppressing basal miR-182–5 p in HepG2.

miR-182–5 p might be a potential biomarker for T2D in obesity, we measured serum concentrations in a subset of n=30 obese ND and n=29 T2D subjects where non-hemolytic serum was available. Serum miR-182–5 p was neither altered in T2D (*Figure 2—figure supplement 1A*) nor correlated with hepatic miR-182–5 p expression (*Figure 2—figure supplement 1B*).

Among the predicted target genes of miR-182–5 p available in miRTarBase and TarBase are genes involved in lipid metabolism and localization (*DHCR7, HMGCS1, INSIG1, SCD, FASN, ELOVL6, ACACA, PNPLA2, PPARA, LRP6*) and glucose homeostasis (*PDK1, INSR, IRS1, FOXO1, G6PC*) (*Grimson et al., 2007*; *Huynh et al., 2011*). According to gene ontology, these genes are involved in T2D-related pathways (*Supplementary file 1h*). Additionally, to identify previously unknown target genes of miR-182–5 p we developed a prediction tool that we called *miRNA-Nvis-tool* which is based on the identification of various potential mRNA-binding sites and favorable binding conditions such as a high AU-content in the surrounding sequence, a non-central location within the 3'-UTR and additional base pairing on the 3' end of the seed (*Friedman et al., 2009*; *Dambal et al., 2015*). Accordingly, we identified *PCK1* as a novel potential target of miR-182–5 p (*Supplementary file 1g*). Using the miRNA Network visualization tool we could confirm the 3'UTR seed match for all database targets except for *ACACA*. Interestingly, only eight of the 16 predicted target genes of miR-182–5 p had previously been validated using cell lines (*Supplementary file 1i*). *ELOVL6* and *LRP6* were previously non-significantly correlated with the expression of hsa-miR-182–5 p in liver metastasis (*Leti et al., 2015*). To corroborate and extend these findings to humans, we measured the hepatic expression of T2D-related miR-182–5 p target genes in our full cohort. The expression of *LRP6, IRS1, PNPLA2, PPARA, PCK1,* and *FOXO1* was significantly (#P<0.05) reduced in T2D compared to ND controls, whereby *LRP6* and *IRS1* were the only targets passing adjustment for multiple testing (*q<0.05; ***q<0.001) (*Figure 2C, Figure 2—figure supplement 1C*). The hepatic expression of *LRP6, IRS1, PPARA,* and *PCK1* correlated negatively with hepatic miR-182–5 p levels (*Figure 2B*, green box and *Figure 2—figure supplement 1D–G*) and with at least one parameter of lipid and glucose homeostasis (*Figure 2B*, red box). *ACACA* correlated with hepatic miR-182–5 p but was not associated with any other parameter. We also observed an intercorrelation between target genes. Reduced *LRP6* was associated with elevated *SCD* and *HMGCS1* expression and to decreased IRS1 expression (*Figure 2B*).

To validate the regulation of target genes by miR-182–5 p, we transfected HepG2 cells with a miR-182–5 p mimic to overexpress miR-182–5 p (*Figure 2—figure supplement 1*). miR-182–5 p overexpression significantly increased the expression of *HMGCS1, SCD, FASN,* and *PNPLA2* (*q<0.05) and reduced the expression of *INSIG1, LRP6, FOXO1* (*q<0.05) and *INSR* (#p<0.05; *Figure 2D*). This

pattern of miR-182–5 p target gene regulation in HepG2 cells partly mirrored the expression pattern observed in livers of T2D subjects. In both conditions, expression of *SCD* and *HMGCS1* was upregulated. Nevertheless, *ELOVL6* and *PNPLA2* showed an opposite trend and *INSR* and *FASN* showed a less pronounced effect in human liver (*Figure 2C*). *LRP6* was the only target gene that was consistently repressed in human liver and in HepG2 cells after miR-182–5 p mimic transfection. Consequently, LRP6 protein levels were significantly decreased after miR-182–5 p overexpression in HepG2 cells (*Figure 2E* and *Figure 2—figure supplement 2*) and renilla luciferase intensity reporting *LRP6* expression was significantly repressed upon miR-182–5 p overexpression in comparison to negative control (nc) transfected cells (*Figure 2F*, left panel). However, when we mutated the central cytosine of the seed sequence into an adenosine (5'-TTG**C**CAA to 5'-TTG**A**CAA) *LRP6* expression was unaltered thus confirming sequence specificity of miR-182–5 p/LRP6 pairing (*Figure 2F*, right panel). Lastly, miR-182–5 p overexpression significantly decreased glucose uptake (*Figure 2G*) and suppressed insulin signaling in HepG2 cells (*Figure 2H* and *Figure 2—figure supplement 3*). Reducing basal miR-182–5 p in HepG2 cells by an antagomir did not change glucose uptake (*Figure 2—figure supplement 4A* and B).

Overall, these results suggest that miR-182–5 p might enhance hepatic de novo lipogenesis via repression of LRP6 and disinhibition of *HMGCS1*, *SCD*, and *FASN*. Furthermore, it decreases hepatic insulin sensitivity by reducing *IRS1* and *INSR* mRNA in obese T2D subjects.

## Hepatic miR-182-5p expression in obese mice can be reversed by weight-loss

To corroborate that hepatic miR-182–5 p plays a role in the regulation of glucose homeostasis and hepatic lipid metabolism, we performed a weight cycling experiment in DIO, glucose intolerant male C57BL/6 J mice (*Figure 3A*). Hepatic miR-182–5 p expression was significantly increased in obese mice exposed to HFD for 24 weeks compared to lean, chow-fed controls (1.9-fold, *Figure 3B*), while 12 weeks of HFD feeding were not sufficient to induce miR-182–5 p expression (data not shown), indicating that miR-182–5 p induction requires chronic metabolic stress and is not a marker of mere obesity. Upon weight-loss by nutritional intervention, i.e., DIO mice were switched from HFD to chow (HC) which equates to mild calorie restriction, hepatic miR-182–5 p levels decreased by 30% demonstrating that the hepatic overexpression of miR-182–5 p in obese mice is reversible. However, hepatic miR-182–5 p expression increased to DIO levels again when the group of mice undergoing weight loss (HC) were re-exposed to HFD to regain all the previously lost weight (YoYo) (*Figure 3B*, *Figure 3—figure supplement 1A*). In case of the other human-murine conserved metabolic miRNAs, only mmu-miR-149–5 p followed this pattern. Other miRNAs reduced expression after HFD feeding and were only partially reversible (*Figure 3—figure supplement 1B*).

Next, we measured the 16 miR-182–5 p target genes in our murine weight cycling model. The hepatic expression of *Lrp6*, *Insr*, *Irs1*, *Ppara*, and *Pck1* was significantly reduced in HFD-fed mice compared to chow controls (*Figure 3C*) which is consistent with the increased expression of miR-182–5 p and the regulation of these target genes - except for *INSR* - in obese T2D human livers. However, whereas *INSR* was not changed in human liver it was significantly decreased in HepG2 cells after miR-182–5 p mimic transfection. Upon weight loss, target gene expression was partially equalized or improved in comparison to chow mice and in comparison to the HFD group the expression of some downregulated miR-182–5 p target genes e.g., *Pck1* was at least partially reversed. Upon weight re-gain, miR-182–5 p expression increased in the YoYo mice (*Figure 3B*) and expression of the targets *Lrp6*, *Irs1* and *Pck1* was significantly repressed (*Figure 3C* and *Figure 3—figure supplement 1C*).

The correlation analysis confirmed the strong associations of miR-182–5 p with *Pck1*, *Irs1*, and *Lrp6* in murine liver (*Figure 3D*, green box) as observed for the human cohort. Furthermore, we could validate the miR-182–5 p associated decrease of *Insr* and *Foxo1* expression, which was previously only observed in HepG2 cells. Metabolic traits such as body weight, plasma leptin and insulin as well as hepatic triacylglycerol (TAG) levels had the strongest correlations with hepatic miR-182–5 p expression in obese mice also in regard to other conserved metabolic miRNAs (*Figure 3D*, red box and *Figure 3—figure supplement 1D*). Taken together, the dynamic regulation of miR-182–5 p and its conserved impact on key genes of insulin signaling and hepatic lipid metabolism in mice and humans (*Figure 3—figure supplement 1E*) highlight the important role of hepatic miR-182–5 p in T2D.

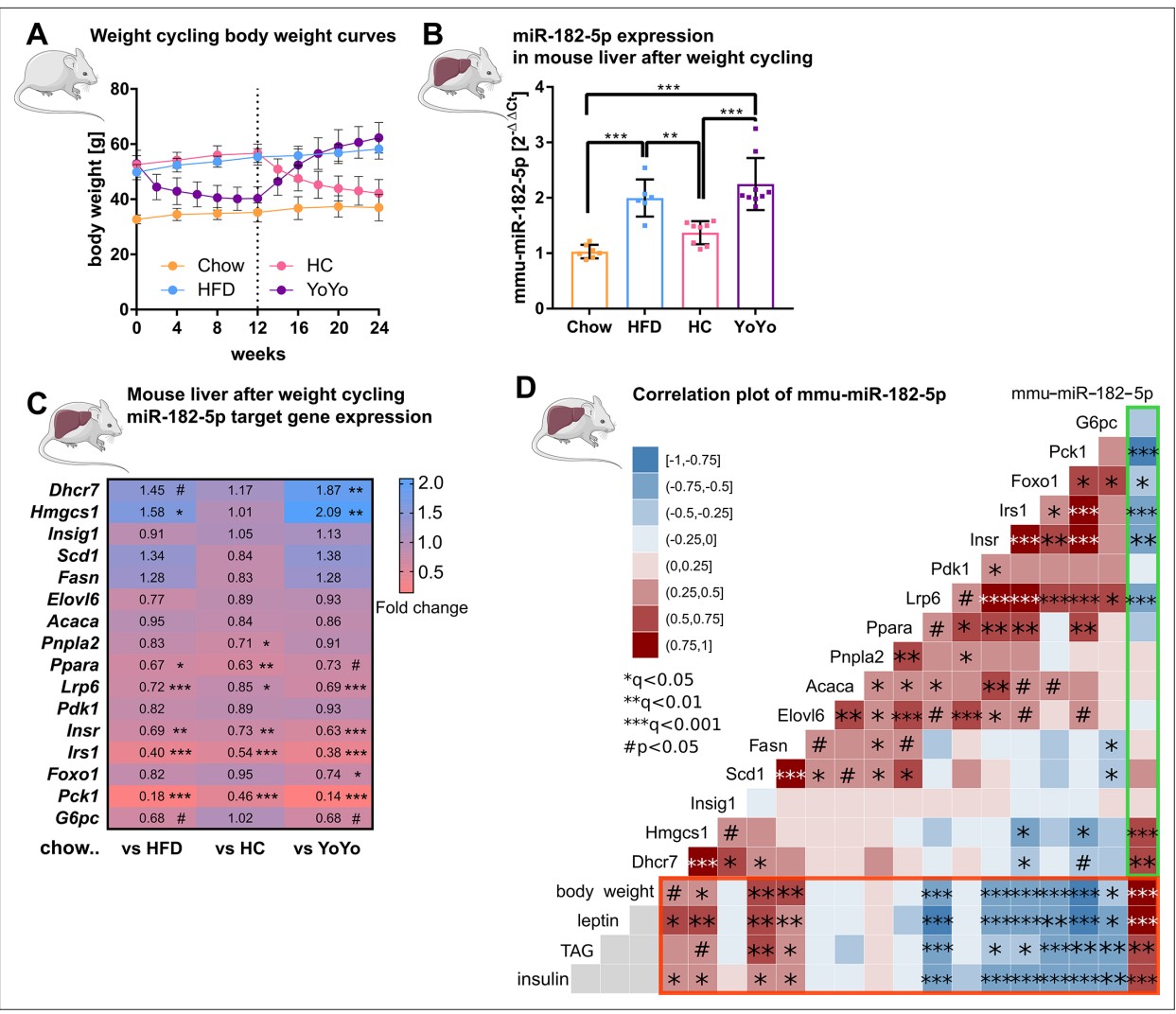

**Figure 3.** Hepatic miR-182–5 p expression in obese mice can be reversed by weight-loss. (**A**) Body weight in grams of mice in the weight cycling cohort (n=7 Chow, n=6 HFD, n=8 HC, n=9 YoYo). Groups of obese high fat diet (HFD)-fed mice were switched to chow for 12 weeks (HC) to induce body weight loss and re-fed with HFD for another 12 weeks (YoYo). (**B**) Expression of miR-182–5 p in liver of mice undergoing weight cycling. (**C**) Expression of target genes of miR-182–5 p in murine liver compared to the Chow control group (see also Figure S2). (**D**) Correlation plot of hepatic miR-182–5 p expression and its target genes in murine liver (green box) and of gene expression with relevant metabolic parameters (red box). Non-tested correlations are indicated by gray squares. TAG: triacylglycerol. Data are shown as mean ± SD (**A,B**) or correlation matrices (**C,D**). Corrected for multiple testing: ***q<0.001, ***q<0.01, *q<0.05, significant prior to adjustment: #p<0.05 (**C, D**); One-Way ANOVA (**B**) or Pearson's correlation (**C,D**).

The online version of this article includes the following source data and figure supplement(s) for figure 3:

**Source data 1.** Phenotypic and qPCR expression (dCT values) data of the weight cycling mouse cohort.

**Figure supplement 1.** Expression and phenotypic association of conserved metabolic microRNAs and target genes in murine liver during weight cycling with a comparison between predicted murine and human regulatory gene networks.

## miR-182-5p overexpression increases hepatic fat and insulin content in metabolically challenged mice

To validate the metabolic effects of miR-182–5 p in vivo, we overexpressed miR-182–5 p in male C57BL/6 J mice by intravenous injections (twice in seven days) of miR-182–5 p mimic or control while receiving HFD (*Figure 4A*). Hepatic miR-182–5 p expression increased 584-fold in mimic treated mice compared to the control group (*Figure 4B*) and was highest expressed in liver compared to spleen and heart (*Figure 4—figure supplement 1A*). Fat mass tended to be increased (p=0.17, *Figure 4C*) whereas body weight remained unchanged. (*Figure 4D*, *Figure 4—figure supplement 1B*). Glucose tolerance (*Figure 4E*, *Figure 4—figure supplement 1C*) and fasting glucose levels (*Figure 3F*) were

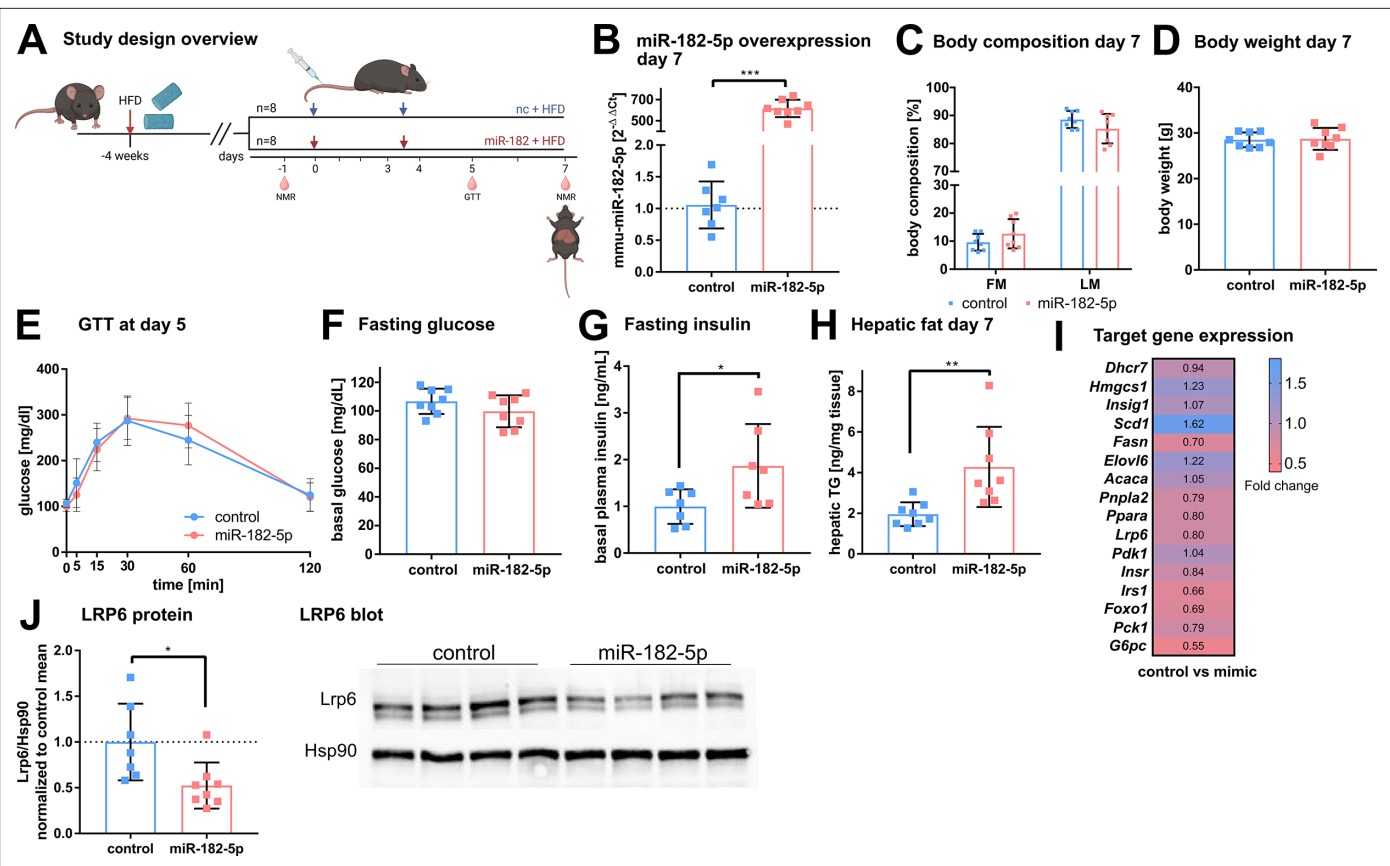

**Figure 4.** In vivo overexpression of miR-182–5p elevates fasting insulin levels and hepatic fat content and reduces LRP6 protein levels. (**A**) Mice were challenged with high fat diet (HFD) for four weeks prior to the first injection of 1 mg miRNA mimic per kg body weight and sacrificed after seven days (n=8 per group if not stated otherwise). A second injection was performed after 3.5 days. Body composition was examined at days –1 and 7 by NMR. Glucose and insulin levels were determined from blood at days –1, 5, and 7. Glucose tolerance was evaluated at day 5. (**B**) Hepatic miR-182–5 p expression was 584-fold upregulated at day 7. (**C**) Fat mass tended to be increased in the miR-182–5 p injected group but (**D**) body weight was not different between control and miR-182–5 p treated mice. (**E**) Glucose tolerance and (**F**) fasting glucose were not different between both groups. (**G**) Fasting insulin levels (n=7 per group) and (**H**) hepatic triglyceride content were increased 2.25-fold and 2.19-fold in miR-182–5 p treated mice, respectively. (**I**) Comparable mRNA, but (**J**) 0.52-fold diminished LRP6 protein levels in liver samples of miR-182–5 p treated mice. The LRP6 blot depicts an example blot of n=4 per group. Data are shown as mean ± SD (**B–H**). *p<0.05, **p<0.01; Student's t-test (**B–H**). Panel A created with BioRender.com, and published using a CC BY-NC-ND license with permission.

The online version of this article includes the following source data and figure supplement(s) for figure 4:

**Source data 1.** Original file for the Western blot analysis of murine LRP6 after treatment with mimic.

**Source data 2.** Phenotypic and qPCR expression (dCT values) data of the mouse trial cohort.

**Figure supplement 1.** Additional phenotypic characteristics of mice overexpressing of miR-182–5 p in liver.

**Figure supplement 1—source data 1.** PDF containing *Figure 4J* highlighted bands and sample labels for protein quantification.

not altered after 5 days of miR-182–5 p overexpression, but fasting insulin levels were significantly increased in miR-182–5 p treated mice (*Figure 3G*) indicating the development of impaired glucose homeostasis. Moreover, miR-182–5 p overexpression significantly increased hepatic triglyceride levels (*Figure 4H*) and lipid droplets (*Figure 4—figure supplement 1D*) which is in line with the trend towards increased fat mass (*Figure 4C*) and our hypothesis based on the findings in obese humans and mice that hepatic miR-182–5 p overexpression would worsen glucose homeostasis and liver fat content. The hepatic expression of *Lrp6, Irs1 FASN, Foxo1,* and *G6pc* tended to be decreased in miR-182–5 p mimic treated mice (*Figure 4I, Figure 4—figure supplement 1E*). Importantly, hepatic LRP6 protein content decreased significantly in mice overexpressing miR-182–5 p (*Figure 4J*) which confirms

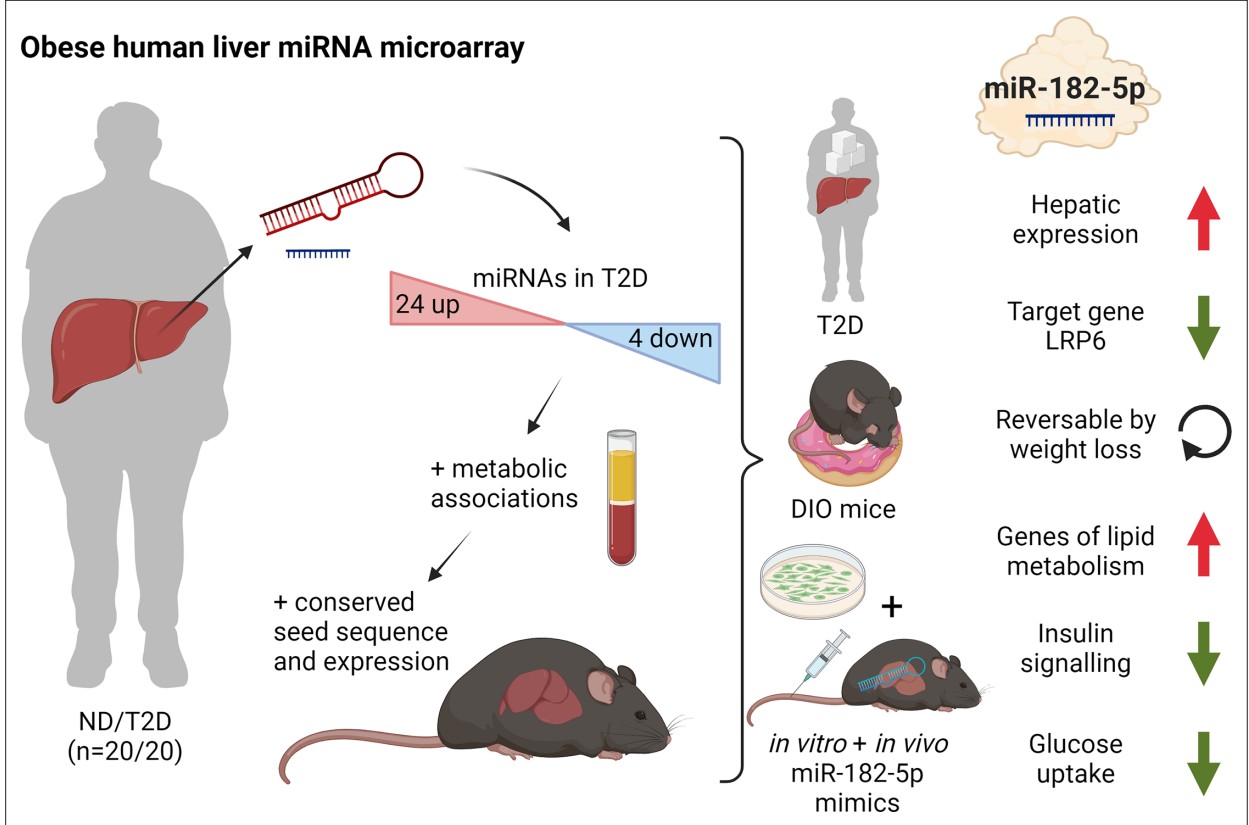

**Figure 5.** Dynamic regulation of insulin resistance and hepatic lipogenesis by miR-182–5 p. Expression of 28 microRNAs was significantly altered in the liver of humans with obesity and type 2 diabetes (T2D) versus humans with obesity but no diabetes (ND). These regulated microRNAs were compared to the hepatic microRNA transcriptome of obese mice and associated with metabolic traits. From those miR-182–5 p was selected as overlapping candidate. LRP6 is the main target gene and consistently altered in humans, cell culture, and obese mice. miR-182–5 p is induced by long-term feeding of high-fat diet in mice and reversed by weight loss. Liver-specific upregulation of miR-182–5 p in metabolically challenged mice induced hepatic fat accumulation and increased fasting insulin levels. Figure created with BioRender.com, and published using a CC BY-NC-ND license with permission.

our data from mimic-treated HepG2 cells and the predicted interaction between miR-182–5 p and Lrp6. Overall, the results from this acute overexpression study indicate that miR-182–5 p has a strong potential to worsen hepatic steatosis and possibly promotes insulin resistance and hyperinsulinemia via downregulation of LRP6.

## Discussion

In the present study, we describe a functional role for the miRNA miR-182–5 p and its target gene *LRP6* in the regulation of glucose homeostasis and hepatic lipid metabolism in obese T2D humans. In a medium-sized but well characterized human cohort, we were able to assess for the first time the hepatic miRNA transcriptome of obese individuals with T2D compared to obese control subjects. We identified a distinct signature of 28 hepatic miRNAs that were associated with T2D of which 19 miRNAs were associated with further metabolic traits. Of those, the human/murine conserved miRNA miR-182–5 p showed the strongest upregulation in T2D subjects and moreover correlated to multiple metabolic traits and pathways of glucose and lipid metabolism including fasting glucose, HbA1c, and liver fat content. The newly validated miR-182–5 p target gene *LRP6* was consistently downregulated in livers of obese T2-diabetics and in obese insulin-resistant mice as well as by in vivo and in vitro miR-182–5 p overexpression. While hepatic miR-182–5 p overexpression increased liver fat content and fasting insulin levels in mice, weight loss in DIO mice reversed miR-182–5 p induction

and partially restored its target genes in liver (*Figure 5*) indicating a therapeutic potential for future miR-182–5 p-based drugs.

Surprisingly, the here presented human hepatic miRNA transcriptome revealed that hepatic T2D-driving miRNAs previously identified in laboratory rodents (*Trajkovski et al., 2011*; *Li et al., 2009*; *Yang et al., 2015*; *Yang et al., 2016*) or liver cell lines (*Gerin et al., 2010*), such as miR-107 (*Trajkovski et al., 2011*) or miR-802 (*Kornfeld et al., 2013*) are likely not dysregulated in obese T2D humans. Only five miRNAs that were dysregulated in livers of obese T2D humans were conserved in obese insulin-resistant mice. Of note, the discrepancy between the here identified miRNAs and known miRNAs regulating glucose metabolism in murine studies (*Trajkovski et al., 2011*; *Kornfeld et al., 2013*; *Yang et al., 2012*; *Yang et al., 2016*; *Dávalos et al., 2011*; *Hanin et al., 2018*) may be based on the design of our human study. Both groups of our human study cohort were severely obese, and our analysis did not compare the obese miRNA transcriptome to a lean control group. Consequently, obesity-induced miRNAs likely escaped our analysis. Furthermore, in our logistic regression analysis, we used the NAFLD activity score (NAS) as cofactor to exclude hepatic lipid content and fibrosis as driving factor for altered miRNA expression. Accordingly, our previously identified T2D-induced hsa-let-7e-5p (*Krause et al., 2020*) shows under this stratification only a strong trend (p=0.0518) to be elevated in T2D. If we exclude NAS as cofactor, we can reproduce the significant association (p=0.0265, *Supplementary file 1b*) which is accompanied by an aging effect (p=0.0286, *Supplementary file 1c*). Nevertheless, we here comprehensively evaluated all hepatic miRNAs transcribed in humans and identified miRNAs specifically dysregulated in obese T2-diabetic, compared to obese non-diabetic subjects.

hsa-miR-182–5 p is transcribed as part of a highly conserved cluster including miR-183/96/182 whose expression is upregulated in a variety of non-sensory diseases, including cancer, neurological, and auto-immune disorders (*Dambal et al., 2015*). Interestingly, in our cohort, only the expression of miR-182–5 p was increased and associated to T2D and other metabolic traits whereas the other cluster members were not detected in the here investigated biopsies. Our findings are in agreement with a previous study where increased hepatic miR-182–5 p expression was associated with the severity of NAFLD-related fibrosis in a smaller cohort of 30 subjects matched for age, sex, BMI and T2D (*Leti et al., 2015*). Furthermore, increased hepatic miR-182–5 p expression levels were also reported in mouse (*Dolganiuc et al., 2009*) and rat (*Nie et al., 2018*) models of NAFLD. Here, we reproduced and mechanistically extended these findings in liver biopsies of 85 obese subjects and in DIO mice undergoing weight-cycling. In addition to the known associations of miR-182–5 p with hepatic lipid metabolism, we revealed strong correlations to genes and clinical parameters associated with glucose tolerance. This new link of miR-182–5 p to glucose homeostasis is further support by our in vitro findings of reduced glucose uptake and impaired insulin signaling after miR-182–5 p overexpression. Additionally, in vivo miR-182–5 p overexpression not only increased hepatic fat content but also increased fasting insulin levels which itself is a predictor of T2D in humans (*Weyer et al., 2000*).

Since miRNAs can be released into circulation from various tissues, serum miRNAs are potential, minimally invasive biomarkers for tissue health (*Hayes and Chayama, 2016*). However, since most of the previously reported circulating miRNAs which are associated with T2D (*Jones et al., 2017*; *Jiménez-Lucena et al., 2018*) are not among the 28 hepatic miRNAs discovered in our analysis (*Figure 1*), our human hepatic miRNA transcriptome now reveals that serum miRNAs might not be suitable to assess dysregulations of hepatic liver glucose and lipid metabolism. We could also not reproduce the previously described association between T2D and serum miR-182–5 p (*Supplementary file 1A* and B) (*Weale et al., 2020*). The validity of serum miR-182–5 p as biomarker for T2D manifestation (*Weale et al., 2020*) in a disease-duration and treatment-specific manner in T2D (*Weale et al., 2021*) could not be tested in our cohort as disease duration and precise duration and dosages of the diabetes treatment was not available for all study participants. Furthermore, serum miR-182–5 p concentrations might be biased by miR-182–5 p released from dead cells of other tissues than liver (*Szilágyi et al., 2020*) and our analysis did not distinguish between cell-free miRNA and extracellular vesicular miRNAs, potentially impacting the diagnostic potential (*Endzeliņš et al., 2017*). Importantly, recent motif analyses identified that miRNAs might contain liver-specific cell-retention signals, such as 'AGAAC' that prevent their release into circulation (*Garcia-Martin et al., 2022*). This retention signal is present in hsa-miR-182–5 p (UUUGGCAAUGGU**AGAAC**UCACACU) (*Garcia-Martin et al., 2022*) and could explain why we did not find differences in serum miR-182–5 p between our study groups. Furthermore, it strengthens our assumption that miR-182–5 p is particularly relevant for

regulating liver metabolism in an autocrine/paracrine manner. Lastly, our results highlight the risk to miss important disease mechanisms when solely focusing on blood-based epigenetic markers (*Krause et al., 2019*) and the urgent need for tissue-specific resource databases as presented in this study.

According to the miRNA-gene interaction database (*Karagkouni et al., 2018*), the lipogenic target genes of miR-182–5 p *SCD* and *FASN* should be downregulated upon miR-182–5 p induction. However, we hypothesize that the increased expression of key genes of cellular de novo lipogenesis is rather an effect of the miR-182–5 p mediated repression of *LRP6* than a direct miRNA-mRNA interaction. *LRP6* is a Wnt co-receptor (*Thompson and Monga, 2007*; *Tamai et al., 2000*) and is linked to metabolic diseases in humans since several loss-of-function mutations within *LRP6* were shown to cause hypertriglyceridemia (*Go, 2015*; *Mani et al., 2007*), hypercholesterinemia (*Liu et al., 2011*; *Ye et al., 2012*), NAFLD (*Go et al., 2014*) and atherosclerosis (*Mani et al., 2007*; *Keramati et al., 2011*). Loss of *LRP6*-mediated activation of the Wnt/beta-catenin signaling pathway (*Figure 5*) enhances hepatic lipid accumulation by increasing de novo lipogenesis and triglyceride synthesis (*Thompson and Monga, 2007*; *Liu et al., 2011*) which is in line with the observed increased expression of *SCD* in T2D obese humans and *SCD* and *FASN* in HepG2 cells after miR-182–5 p mimic transfection. Additionally, mice carrying the loss-of-function mutation LRP6 p.R611C develop fatty liver disease including insulin resistance, liver inflammation, and steatosis due to increased hepatic de novo lipogenesis (*Wang et al., 2015*). Besides its crucial role as regulator of de novo lipogenesis *LRP6* regulates glucose metabolism by promoting TCF7L2-dependent insulin receptor expression in humans (*Singh et al., 2013*) and by regulating *IRS1* expression in cells (*Bommer et al., 2010*). Our findings of reduced *IRS1* expression in diabetic obese humans and reduced *INSR* and *IRS1* expression in DIO mice as well as downregulated glucose uptake and insulin signaling in miR-182–5 p overexpressing cells are consistent with this glucoregulatory role of *LRP6*, acting under tight control by miR-182–5 p. Our acute hepatic miR-182–5 p overexpression study in mice strongly corroborates this miR-182–5 p-Lrp6 link, and warrants additional, long-term studies on whole body glucose tolerance and hepatic metabolism in miR-182–5 p overexpressing mice.

In naïve mice, hepatic miR-182–5 p expression was only upregulated after 20 weeks of HFD feeding, i.e., long after the mice developed obesity compared to standard diet-fed mice. Accordingly, obesity per se does not seem to be the sole driver of hepatic miR-182–5 p. This is consistent with the human situation, were only diabetic obese but not 'healthy' obese humans show the increase of hepatic miR-182–5 p expression. Whether miR-182–5 p dictates if an obese individual becomes type 2 diabetic or remains glucose tolerant, and whether miR-182–5 p mechanistically drives hepatic dysregulation of glucose and lipid metabolism, are thus intriguing questions for future studies. Such follow-up studies should be conducted in humans, given that mice are rarely developing diabetic pathologies surpassing glucose intolerance, and should compare lean with overweight or obese non-diabetic and T2D subjects to specifically address whether exacerbated hepatic insulin resistance and lipid deposition affect miR-182–5 p expression levels. In our cohort, T2D subjects received anti-diabetic treatment, which is a limitation of our study. HbA1c levels in the T2-diabetic subjects were nonetheless still significantly higher compared to the ND obese individuals (*Supplementary file 1a*). To ultimately dissect if hepatic miR-182–5 p upregulation is causally involved in the development of insulin resistance or if it is a marker of manifested T2D would mandate longitudinal studies. First mechanistic evidence is revealed by our in vitro and in vivo studies, which show that miR-182–5 p overexpression acutely decreases glucose uptake and insulin signal transduction in hepatic cells and increases fasting insulin levels and hepatic fat content in mice. Future studies should investigate the mechanisms inducing miR-182–5 p expression in the liver of T2-diabetic subjects, and whether miR-182–5 p antagonism can rescue the metabolic phenotype in mice.

## Conclusions

We provide a network of novel microRNAs that are dysregulated in livers of T2-diabetic subjects and identify miR-182–5 p and its target genes as potential drivers of dysregulated glucose tolerance and fatty acid metabolism in obese T2-diabetics. Mechanistic studies with miRNA mimics in cells and mice revealed that hepatic miR-182 expression elicits repressive effects on its target gene *LRP6* and subsequently on glucose and lipid metabolism. The discoveries that miR-182–5 p is associated with T2D and NAFLD via *LRP6* downregulation in liver of obese subjects and the reversibility by weight loss in DIO mice offer unique mechanistic insight into diabetes pathology and point towards promising

future anti-diabetic strategies, built either on antagonizing miR-182–5 p activity or reducing hepatic miR-182–5 p expression by pharmacological or dietary means.

## Acknowledgements

We thank M Grohs (Institute for Human Genetics, University of Lübeck), P Schroeder and A Heinicke (both Department of General, Visceral and Thoracic Surgery, University Medical Center Hamburg-Eppendorf) as well as Katrin Huber, Noémi Mallet and Miriam Krekel (Research Unit NeuroBiology of Diabetes, Helmholtz Zentrum Munich) for their excellent technical assistance. Graphics were created with Bio Render (https://biorender.com/).

## Additional information

### Funding

| Funder | Grant reference number | Author |
| --- | --- | --- |
| Deutsche Forschungsgemeinschaft | KI 1887/2-1 | Henriette Kirchner |
| Deutsche Forschungsgemeinschaft | KI 1887/2-2 | Henriette Kirchner |
| Deutsche Forschungsgemeinschaft | KI 1887/3-1 | Henriette Kirchner |
| Deutsche Forschungsgemeinschaft | GRK-1957 | Henriette Kirchner |
| Deutsche Forschungsgemeinschaft | CRC-TR296 | Henriette Kirchner |
| European Research Council | 101002247 | Paul Pfluger |
| Deutsches Zentrum für Diabetesforschung | 82DZD09D1G | Sonja C Schriever Henriette Kirchner |

The funders had no role in study design, data collection and interpretation, or the decision to submit the work for publication.

### Author contributions

Christin Krause, Conceptualization, Data curation, Formal analysis, Investigation, Methodology, Writing - original draft, Writing - review and editing; Jan H Britsemmer, Data curation, Investigation, Methodology; Miriam Bernecker, Anna Molenaar, Natalie Taege, Nuria Lopez-Alcantara, Cathleen Geißler, Katharina Iben, Anna Judycka, Investigation, Methodology; Meike Kaehler, Formal analysis, Methodology; Jonas Wagner, Stefan Wolter, Investigation, Project administration; Oliver Mann, Resources, Project administration; Paul Pfluger, Hendrik Lehnert, Resources, Funding acquisition, Writing - review and editing; Ingolf Cascorbi, Resources; Kerstin Stemmer, Conceptualization, Resources, Methodology; Sonja C Schriever, Conceptualization, Funding acquisition, Investigation, Methodology, Writing - review and editing; Henriette Kirchner, Conceptualization, Supervision, Funding acquisition, Investigation, Writing - original draft, Project administration, Writing - review and editing

### Author ORCIDs

Christin Krause ⓘ http://orcid.org/0000-0001-8703-0023
Miriam Bernecker ⓘ http://orcid.org/0009-0004-8324-7515
Ingolf Cascorbi ⓘ http://orcid.org/0000-0002-2182-9534
Sonja C Schriever ⓘ http://orcid.org/0000-0001-5461-3214
Henriette Kirchner ⓘ http://orcid.org/0000-0002-7887-247X

### Ethics

All participants of the human cohort signed informed consent. The study was approved by the local ethics committee (Ärztekammer Hamburg PV4889) and conformed to the ethical guidelines of the 1975 Declaration of Helsinki.

The murine studies were approved by the State of Bavaria, Germany.(approval ID ROB-55.2-2532. Vet_02-19-167).

Reviewer #1 (Public review): https://doi.org/10.7554/eLife.92075.3.sa1
Reviewer #2 (Public review): https://doi.org/10.7554/eLife.92075.3.sa2
Reviewer #3 (Public review): https://doi.org/10.7554/eLife.92075.3.sa3
Author response https://doi.org/10.7554/eLife.92075.3.sa4

## Additional files

### Supplementary files

• Supplementary file 1. Clinical Characteristics for the complete (n=85) and the microarray (n=40) human liver cohort. A rank sum test was performed to find significant differences between both subgroups.

• Supplementary file 2. Primer sequences for SYBR-green analysis of cDNA by qPCR and cloning.

• Supplementary file 3. Assay-IDs for TaqMan Gene-expression Assays and TaqMan Advanced miRNA Assays.

• Supplementary file 4. Associations between miRNAs and type 2 diabetes (T2D). Logistic regression models for the incidence of T2D were generated using self-written scripts in MATLAB. Age, sex, BMI, and the NAFLD activity score (NAS) were used as additional cofactors. The coefficient estimate and p-value are indicated for the respective miRNA expression. ID is the respective identifier for the human microRNA on the GeneChip miRNA 4.0 Array.

• Supplementary file 5. Associations between miRNAs and metabolic traits. Effect sizes describe the change of the trait if the miRNA expression changes by 1 log2 value. Age, sex, BMI, and the NAFLD activity score (NAS) were used as additional cofactors whenever they did not serve as response value. ID is the respective identifier for the human microRNA on the GeneChip miRNA 4.0 Array.

• Supplementary file 6. Associations between miRNAs and excluding factors age and BMI. To control for associations with both confounding factors, also the HbA1c level was considered as cofactor for linear regression models. Effect sizes describe the change of the trait if the miRNA expression changes by 1 log2 value. ID is the respective identifier for the human microRNA on the GeneChip miRNA 4.0 Array.

• Supplementary file 7. Predicted target genes from miRTarBase used for pathway analysis.

• Supplementary file 8. Candidate genes for hsa-mR-182–5 p from database entries stratified by metabolic pathways from Gene Ontology. Additional column qPCR contains target genes which were considered for analysis by qPCR in the complete cohort.

• Supplementary file 9. Result of all consulted target gene databases and a comprehensive 3'UTR screening for each potential target gene of hsa-miR-182–5 p. Besides the relative position of the seed sequences within the 3'-UTR of the target gene, also the relative AU content of the surrounding 60 nt, whether there is an additional base pairing between the mRNA, and the miRNA 3' of the seed and the relative position of the seed within the target 3'UTR is listed.

• MDAR checklist

• Reporting standard 1. STROBE checklist for cross-sectional studies.

### Data availability

Sequencing data have been deposited in GEO under accession codes GSE176025 (human data) and GSE211367 (mouse data). Source data for Figures 2, 3 and 4 have been provided. Source code for the target gene prediction with the miRNA Network visualization tool, together with a database (SQLite) with all association parameters, is available at GitHub (https://github.com/christinkrause55/microRNA_network_visualizer; copy archived at *Krause, 2024*).

The following datasets were generated:

| Author(s) | Year | Dataset title | Dataset URL | Database and Identifier |
|---|---|---|---|---|
| Krause C, Geißler C, Iben K, Grohs M, Kaehler M, Wagner J, Wolter S, Mann O, Cascorbi I, Lehnert H, Kirchner H | 2023 | Liver microRNA transcriptome reveals miR-182 as link between type 2 diabetes and fatty liver disease in obesity | https://www.ncbi.nlm.nih.gov/geo/query/acc.cgi?acc=GSE176025 | NCBI Gene Expression Omnibus, GSE176025 |
| Krause C, Britsemmer JH, Pfluger PT, Kirchner H, Schriever SC | 2023 | microRNA expression data from liver samples of mice fed with a high fat diet for 24 weeks | https://www.ncbi.nlm.nih.gov/geo/query/acc.cgi?acc=GSE211367 | NCBI Gene Expression Omnibus, GSE211367 |

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
