## [Editor Report · eLife assessment]

Building on on the observation of an increase in miR-182-5p in diabetic patients, the authors investigated the role of miR-182-5p and its target gene LRP6 in dysregulated glucose tolerance and fatty acid metabolism in obese type 2 diabetics. The use of human livers complemented by supporting data in mice and cells are strengths, but the evidence presented remains **incomplete**. The findings provide **valuable** insights into the role of miRNAs in the regulation of liver metabolism and insulin sensitivity in individuals with diabetes and fatty liver disease.

---

## [Referee Report · Reviewer #1 (Public review)]

Summary:

This study demonstrated a novel exciting link between conserved miRNA-target axis of miR-182-Lrp6 in liver metabolism which causatively contributes to type 2 diabetes and NAFLD in mice and, potentially, humans.

Strengths:

The direct interaction and inhibition of Lrp6 by miR-182 is convincingly shown. The effects of miR-182-5p on insulin sensitivity are also credible for the in vivo and in vitro gain-of-function experiments.

Weaknesses:

However, the DIO cohorts lack key assays for insulin sensitivity such as ITT or insulin-stimulated pAKT, as well as histological evidence to support their claims and strengthen the link between miR-182-5p and T2D or NAFLD. Besides, the lack of loss-of-function experiments limits its aptitude as potential therapeutic target.

---

## [Referee Report · Reviewer #2 (Public review)]

Summary:

In this study, Christin Krause et al mapped the hepatic miRNA-transcriptome of type 2 diabetic obese subjects, identified miR-182-5p and its target genes LRP6 as potential drivers of dysregulated glucose tolerance and fatty acid metabolism in obese T2-diabetics.

Strengths:

This study contains some interesting findings and are valuable for the understanding of key regulatory role of miRNAs in the pathogenesis of T2D.

Weaknesses:

The authors didn't systemically investigate the function of miR-182 in T2DM or NAFLD.

---

## [Referee Report · Reviewer #3 (Public review)]

Summary:

In this manuscript, Krause and colleagues identify miR-182 as diabetes-associated microRNA: miR-182 is increased in bariatric surgery patients with versus without T2D; miR-182 was the only microRNA associated with three metabolic traits; miR-182 levels were associated with increased body weight in mice under different dietary manipulations; overexpression in Hep-G2 led to a decrease in LRP6; and overexpression in HFD fed mice led to increased insulin and liver TG. The manuscript provides a potentially useful resource of microRNA expression in human livers, though the functional importance of miR-182 remains unclear.

Strengths:

The use of human tissues and good sample sizes is strong.

Weaknesses:

The study remains primarily correlative; the in vivo overexpression is non-physiological; and the mechanisms by which miR-182 exerts its effects are not rigorously tested.

---

## [Author Response]

The following is the authors’ response to the original reviews.

**Reviewer #1 (Recommendations For The Authors):**
(1) Original blots in Figures 2E and 2H should be shown as well as the quantification of miR-182-5p overexpression in HepG2 cells. miR-182-5p expression in T2D patients was 2.3-fold higher than ND patients. The lack of insights into the degree of miR-182-5p overexpression precluded proper interpretation of the data presented.

Thank you very much for these comments. We now include the original uncut blots and relevant bands (new supplementary figure 3A) as well as the quantification of miR-182-5p expression in mimic-treated HepG2 cells in the supplement (new supplementary figure 2).

(2) What are the upstream transcriptional regulators of miR-182-5p?

To the best of our knowledge the upstream transcriptional regulators of miR-182-5p are currently unknown.

(3) What's the purpose of the weight cycling cohort? Figure 3A only showed that miR-182-5p expression was highly correlated to body weight, but the cohort can not explain why the human cohort has different miR-182-5p expression. GTT and ITT data are lacking for this cohort and thus cannot demonstrate a causal link between insulin sensitivity and miR-182-5p. The lack of histological evidence cannot show the relationship between NAFLD and miR-182-5p.

The purpose of the weight cycling cohort was to demonstrate that miR-182-5p is dynamically altered and that it can be reversed to almost control levels by weight loss. Thereby we validate in mice that obesity is associated with miR-182-5p upregulation (HFD group without intervention) and we propose that the adverse effects of increased miR-182-5p in obesity might be reversible by weight loss. We did not perform ITTs and GTTs in this weigh cycling cohort because the HFD-model in C57BL/6 mice is well established and it can be assumed that glucose- and insulin-tolerance deteriorated during HFD feeding (doi.org/10.1038/oby.2007.608; doi:10.1007/978-1-61779-430-8_27) and improved after weight loss (doi:10.1038/s41598-023-40514-w). To corroborate this assumption, we provide plasma insulin along with as other important metabolic marker of the weight cycling model in supplemental figure 5A.

(4) Loss-of-function of miR-182-5p and/or gain-of-function of Lrp6 in vivo or in vitro would clarify the importance of the miR-182-5p-Lrp6 axis and provide more direct evidence for its potential as a therapeutic target.

We absolutely agree with the reviewer that loss of miR-182 and gain of LRP6 function experiments are missing. However, we provide miR-182 gain of function experiments that impressively show increased liver triglycerides after only seven days of miR-182 overexpression. Because these in vivo data are only short-term, we stated our conclusions carefully and point out that we do not have evidence for a direct involvement of miR-182-5p in insulin signaling. We are now planning follow-up studies in which miR-182-5p will be overexpressed and also antagonized for a longer time. However, for the timeframe of this revision process these extensive studies are not feasible and we ask the reviewer for his/her understanding.

(5) The schematic summary is too complex and includes too many assumptions to faithfully represent the data shown in this study.

We agree, the schematic summary is very complex. Therefore we simplified the upper part (new figure 5) and only focused on the clearly regulated genes and main pathways.

**Reviewer #2 (Recommendations For The Authors):**
(1) Although lots of microarray analyses were performed in this study, the authors didn't systemically investigate the function of miR-182 in T2DM or NAFLD. The current data provided in this manuscript may only support that miR-182 is involved in the homeostasis of glucose or insulin.

We thank the reviewer for this comment and agree that the nature of or data is mostly correlative. We tried to overcome this by performing mechanistic in vitro data. Because overexpression of miR-182-5p decreases inulin signaling in vitro and induces hyperinsulinemia in vivo we still strongly believe that miR-182-5p is highly relevant for the homeostasis of glucose and insulin.

(2) The authors used miRNA mimics to overexpress miR-182 in mice. How to emphasize the target specificity in the liver? Normally, adeno-associated virus 8 (AAV8) is used to specifically target the liver.

Tail vein injections as used in our experimental set-up are known to deliver compounds directly to the liver via the portal vein. For modulation of microRNAs in the liver it is an established technique to deliver mimics (or inhibitors) via the tail vein (doi:10.1007/978-1-62703-435-7_18; doi: 10.1089/10430349950017734). To account for off-target effects we quantified miR-182-5p and target gene expression in spleen and heart. Although miR-182-5p concentrations in mimic treated mice were strongly increased in these tissues, expression in the liver was still highest (new supplementary figure 6A).

(3) The HE and Oil red staining of the mouse liver should be shown in miR-182-5p overexpressing mice compared with the control mice, which could provide a more intuitive view of the fat content in the mouse liver.

Unfortunately the livers were flash frozen and not optimally prepared for later histological analyses. Nevertheless, we performed H&E stainings in all livers and provide representative HE stainings of two control and two miR-182-mimic treated mice (new supplementary figure 5D). The increase hepatic lipid content is clearly visible in the H&E staining of miR-182-mimic treated mice and supports our previous findings of increased hepatic triglycerides (Figure 4H). Due to the freezing process, livers were damaged and Oil red staining was impossible.

(4) After overexpression of miR-182-5p in mice, the serum insulin levels were increased. Does miR-182-5p affect insulin resistance in mice? The insulin tolerance test (ITT) experiment needs to be performed.

We thank the reviewer for this comment. Indeed, the performance of an ITT would have clarified the effects of miR-182 on insulin tolerance best. Because we did not see differences in the GTT after treating mice acutely with the miR-182 mimic we decided to not perform the ITT in this short-term. The increased fasting serum levels after miR-182-5p mimic treatment (Fig. 4G) suggest that rather insulin sensitivity than insulin secretion is disturbed by miR-182-5p. We are aware, that in future experiments mice should be treated for a longer period with miR-182-5p mimics and that an ITT should be performed in these more chronic studies.

(5) In Figure 2H, the author measured the level of p-Akt/Akt to indicate the effect of miR-182-5p on insulin resistance in HepG2 cells. It is best to provide the western blotting results of p-AKT and t-AKT after HepG2 cells are treated with or without insulin.

We now provide the full blots for all western blotting experiments as new supplemental figure 3B. The HepG2 cells were stimulated with 20 nM insulin 10 min before harvest as described in 2.11 and consequently Akt and p-Akt were quantified. We did not analyze Akt and p-Akt without stimulation because Akt is rarely phosphorylated in the basal non-insulin stimulated state.

(6) This study suggests that miR-182-5p may promote insulin resistance and hyperinsulinemia by downregulating LRP6. Nevertheless, to confirm this conclusion, we suggest you transfect miR-182-5p after downregulating the level of LRP6 with its siRNA for further validation.

Because miR-182-5p targets LRP6 as we have validated by luciferase-assays, LRP6 levels are already low after miR-182-5p overexpression. Thus, the additional downregulation of LRP6 by other means (such as siRNAs) does not make sense in our opinion.

(7) The author described that serum miR-182-5p was neither altered in T2D nor correlated with hepatic miR-182-5p expression, so is it suitable as the biomarker of T2D?

Yes, as the reviewer stated correctly, serum concentrations of miR-182-5p were not related to its liver concentrations or the type 2 diabetic state. We therefore suggest that circulating miR-182-5p levels are not a suitable biomarker for T2D. We clarified this in the discussion.

(8) What are the changes in fasting blood glucose levels in HFD, HC, and YoYo mouse models? Is there a correlation between miR-182-5p level and fasting blood glucose level in T2D patients and mouse models?

Unfortunately, we did not measure the fasting blood glucose levels in this mouse model and therefore cannot answer this question. However, we provide the fasting insulin levels of our mouse models and their positive correlations with miR-182-5p (Fig. 3D and Suppl.Fig. 5D). In T2D humans, hepatic miR-182-5p correlates positively with fasting glucose (Fig. 2B).

(9) The capitalization of the letters in "STrengthening the Reporting of OBservational studies in Epidemiology" should be checked. What does the "Among these is miRNAs miR-182-5p" mean? Please clarify it.

The “STrengthening the Reporting of OBservational studies in Epidemiology “ report form is abbreviated as “STROBE” list. We this capitalized the letters that are used to build the abbreviation.

“Among these is miRNAs miR-182-5p” is a typo for which we apologize. It should mean “Among these conserved miRNAs is miR-182-5p.” We corrected this error.

**Reviewer #3 (Recommendations For The Authors):**
(1) The functional importance of miR-182 on gene expression is not rigorously tested.(A) Many of the target genes in Fig. 1C and Fig. 3 are controlled by multiple factors that are known to be increased with obesity (e.g., lipogenic genes are increased by hyperinsulinemia), making it likely that their association with miR-182 is correlative rather than a consequence of miR-182 increases.

We thank the reviewer for this comment and agree that miR-182 is not the only factor regulating the here investigated genes. We rather propose, that miR-182 could be an additional upstream regulator that holds the potential to modify entire pathways of insulin signaling and lipogenesis. However, miR-182 should be not viewed as an on/off-switch as it likely plays a modulating role. Although, our in vivo data stemming from humans and mice are correlative we believe that the in vitro data derived in HepG2 cells clearly show a causal role for miR-182-5ß in decreasing LRP6 and insulin signaling, indicated by lower AKT phosphorylation after miR-182-5p overexpression.

(B) 500-fold overexpression of miR-182 does not significantly change gene expression. The authors need to knockdown miR-182 in mice and then feed them a chow versus high-fat diet. If miR-182 is a significant regulator of these genes, the effects of the diet will be blunted.

We thank the reviewer for the constructive criticism and agree that an optimal experiment would be to antagonize miR-182-5p in mice to rescue glucose and lipid metabolism. There here presented in vivo upregulation of miR-182-5p was a proof-of-concept study to confirm our hypothesis in a reasonable timeframe. We are aware, that follow-up studies are needed, and we are now planning studies in which miR-182-5p will be overexpressed and also antagonized for a longer time. However, for the timeframe of this revision process these extensive studies are not feasible and we ask the reviewer for his/her understanding.

(2) It has previously been shown that miR-182 is in a polycistrionic microRNA locus that is activated directly by SREBP-2. Is this also true in humans? If so, this would indicate that miR-182 is a marker of SREBP activity. How does the nuclear active form of SREBP1 and SREBP2 change in the human livers and HFD-fed mice?

We thank the reviewer for this very interesting question. Suitable experiments to investigate if miR-182-5p is activated by SREBF would be EMSAs or ChIPs. Unfortunately we have only frozen protein lysate of the human livers left in which such experiments cannot be performed. We agree that this should be prioritizes in the future.

(3) Similarly, to test the role of LRP6 in mediating the effects of miR-182, the authors should compare the effects of miR-182 overexpression in the presence and absence of LRP6.

Because miR-182-5p targets LRP6 as we have validated by luciferase-assays, LRP6 levels are already low after miR-182-5p overexpression. Thus, the additional downregulation of LRP6 by other means (such as siRNAs) does not make sense in our opinion.

(4) The methods are a bit confusing. The authors state that "we applied a logistic regression analysis for the 594 mature miRNAs using the NAFLD activity score (NAS) as a cofactor to exclude any bias by hepatic fat content, lobular inflammation, and fibrosis." However, they later showed that miR-182 levels are correlated with NAS. Please clarify.

We excluded NAFLD explicitly as driving factor for the association to T2D by including a surrogate (the NAFLD activity score) as cofactor. It is well known that NAFLD and T2D are indeed likely associated to each other. Since not all our included individuals with T2D have NAFLD and vice versa, a second correlation with NAS revealed also that a high NAS is associated with higher expression of miR-182.

(5) Does two-fold overexpression of miR-182 (which mimics the effects of HFD) have any effect on chow-fed mice?

This is a very interesting question that we unfortunately cannot answer right now. We are planning further mouse studies in which we will include a chow-fed mice as controls.